# CD56 regulates human NK cell cytotoxicity through Pyk2

Justin T Gunesch[1†], Amera L Dixon[1,2,3], Tasneem AM Ebrahim[3,4], Melissa M Berrien-Elliott[5], Swetha Tatineni[2‡], Tejas Kumar[2§], Everardo Hegewisch-Solloa[3], Todd A Fehniger[5], Emily M Mace[3*]

[1]Baylor College of Medicine, Houston, United States; [2]Rice University, Houston, United States; [3]Department of Pediatrics, Vagelos College of Physicians and Surgeons, Columbia University Irving Medical Center, New York, United States; [4]Barnard College, New York, United States; [5]Washington University School of Medicine, St. Louis, United States

*For correspondence:
em3375@cumc.columbia.edu

Present address: †Immatics, Houston, United States; ‡University of Chicago Pritzker School of Medicine, Chicago, United States; §Baylor College of Medicine, Houston, United States

Competing interests: The authors declare that no competing interests exist.

**Abstract** Human natural killer (NK) cells are defined as CD56$^+$CD3$^-$. Despite its ubiquitous expression on human NK cells the role of CD56 (NCAM) in human NK cell cytotoxic function has not been defined. In non-immune cells, NCAM can induce signaling, mediate adhesion, and promote exocytosis through interactions with focal adhesion kinase (FAK). Here we demonstrate that deletion of CD56 on the NK92 cell line leads to impaired cytotoxic function. CD56-knockout (KO) cells fail to polarize during immunological synapse (IS) formation and have severely impaired exocytosis of lytic granules. Phosphorylation of the FAK family member Pyk2 at tyrosine 402 is decreased in NK92 CD56-KO cells, demonstrating a functional link between CD56 and signaling in human NK cells. Cytotoxicity, lytic granule exocytosis, and the phosphorylation of Pyk2 are rescued by the reintroduction of CD56. These data highlight a novel functional role for CD56 in stimulating exocytosis and promoting cytotoxicity in human NK cells.

## Introduction

Natural killer (NK) cells are innate immune effectors that play an important role in the clearance of virally infected and tumorigenic cells and modulation of immune responses. Human NK cells represent approximately 10% of circulating lymphocytes and can be defined within this population as CD56$^+$CD3$^-$ cells. CD56$^{dim}$ cells are the predominant subset in peripheral blood while the minority of the circulating NK cell population are CD56$^{bright}$ cells. CD56$^{bright}$ and CD56$^{dim}$ NK cells have unique expression of cell surface receptors, transcription factors and intracellular effector molecules that contribute to their distinct phenotypic and functional capacities (*Caligiuri, 2008*; *Freud and Caligiuri, 2006*). Despite its conserved expression on human NK cells and its use as a phenotypic identifier, the role of CD56 in immune function is poorly defined.

CD56 is the cluster of differentiation nomenclature for neural cell adhesion molecule (NCAM). In addition to being abundantly expressed in cells of neuronal origin, NCAM is also expressed in other tissues including the heart, kidney, skeletal muscles, liver, and on hematopoietic-derived cells including dendritic cells and human natural killer (NK) and NKT cells (*Galuska et al., 2017*; *Lanier et al., 1989*; *Roothans et al., 2013*). NCAM is a member of the immunoglobulin superfamily of cell adhesion molecules and has multiple isoforms due to RNA splicing (*Cunningham et al., 1987*; *Shapiro et al., 2007*; *Williams and Barclay, 1988*).The most abundant isoforms are the glycophosphatidylinositol (GPI)-anchored 120 kDa form (NCAM120) and two isoforms that contain transmembrane endodomains, NCAM140 and NCAM180 (*Cunningham et al., 1987*; *Kleene and Schachner, 2004*). The extracellular portion of NCAM is highly conserved between the isoforms and contains 5 Ig-domains and two fibronectin-III domains that mediate homophilic (NCAM-NCAM) and

**eLife digest** The immune system deploys different cell types to take out cancer cells. True to their name, one type of immune cell known as natural killer cells kills tumor target cells by releasing toxic proteins that kill the harmful cells. In humans, these immune cells are defined, among other things, by the presence of a protein called CD56 on their cell surface. This protein (which is also known as NCAM) is thought to help cells to stick to their surroundings and control their movements. However, it was not clear whether CD56 also plays a role in the destructive abilities of natural killer cells.

Gunesch et al. have now looked to see what would happen if natural killer cells lacked CD56 on their surface. The experiments included deleting the gene for CD56 from two kinds of human natural killer cell that are commonly grown in the laboratory (called NK92 and YTS). In both cases, the cells lacking CD56 killed fewer cancer cells than the unedited natural killer cells. The NK92 cells were much more affected by the loss of CD56 than the YTS cells, and after Gunesch et al. compared the two kinds of cell they identified another protein called Pyk2 as the potential reason behind the difference.

The Pyk2 protein is known to help a natural killer cell latch onto target cancer cells and release its toxic proteins. To do this, Pyk2 must first be activated with phosphate groups via a process known as phosphorylation. Gunesch et al. showed that Pyk2 protein in unedited NK92 cells was more highly phosphorylated than those of the YTS cells, and that Pyk2 activation by phosphorylation was greatly decreased in NK92 cells when the gene for CD56 was deleted. Together these and other results suggest that CD56 on natural killer cells helps to promote Pyk2 to activate the cells' cancer-killing abilities through Pyk2 phosphorylation, especially in NK92 cells.

These findings open up new lines of investigation into the relationship between sticky surface proteins and the activation of immune cells. They may also have important implications for the use of the immune system to treat cancer via immunotherapy.

heterophilic interactions between cells and the extracellular matrix (*Soroka et al., 2003*; *Walmod et al., 2004*). NCAM$^{-/-}$ mice have impaired learning and memory and behavioral disorders that are attributed to impaired nervous system development and post-differentiation maintenance of signaling and plasticity (*Brandewiede et al., 2014*; *Cremer et al., 1994*). As such, NCAM plays an important role in non-immune cellular differentiation and function.

In addition to its role as an adhesion molecule, NCAM signaling in neuronal cells is important for neurite outgrowth and synaptic plasticity. NCAM140 is constitutively associated with the membrane-associated Src-family tyrosine kinase Fyn in axonal growth cones (*Beggs et al., 1997*). Signaling through NCAM140 by agonist antibodies induces the recruitment of the non-receptor tyrosine kinase focal adhesion kinase (FAK) in neuronal cells, leading to a rise in intracellular Ca++ and neurite outgrowth (*Beggs et al., 1997*; *Ditlevsen et al., 2008*; *Schmid et al., 1999*). In addition to this signaling pathway, NCAM activates fibroblast growth factor receptor −1 (FGFR1) following trans-homophilic binding or binding of soluble extracellular NCAM and activates a non-canonical signaling pathway via phospholipase Cγ (PLCγ) that activates Erk1/2 in a Src-kinase dependent manner (*Francavilla et al., 2009*).

NCAM is unique amongst other glycoproteins in that its fifth Ig-domain contains 2 *N*-linked glycosylation sites that can be highly polysialylated (*Cunningham et al., 1987*; *Kleene and Schachner, 2004*). PSA-NCAM is particularly important for synaptic plasticity and is highly expressed in embryonic development, with decreasing and restricted expression found in adults. The phenotype of olfactory bulb precursor cell migration in NCAM-deficient mice can be recapitulated by enzymatic removal of PSA, demonstrating the importance of PSA in NCAM signaling (*Ono et al., 1994*). The lack of expression of NCAM or a known NCAM homologue on murine NK cells precludes the use of mouse models to test the in vivo requirement for NCAM or PSA-NCAM on NK cells. However, while murine NK cells are not modified by PSA, a subset of murine myeloid precursors in bone marrow are PSA$^{high}$, and mice deficient for the ST8Sia IV sialyltransferase exhibit a sustained and severe contact hypersensitivity response, suggesting a role for PSA in the functional immune response (*Drake et al., 2008*). Further investigation demonstrated that C-C motif chemokine receptor 7

(CCR7) is a carrier of PSA on dendritic cells. Binding of PSA on CCR7 to the C-C motif chemokine ligand 21 (CCL21) relieves CCL21 autoinhibition, leading to DC migration and helping confer specificity for response to chemokine (*Kiermaier et al., 2016*). Together these studies underscore the significance of this unusual protein modification in both neuronal cells and certain immune contexts, however the role of PSA addition to NCAM on human NK cells has not been defined.

NK cell cytotoxic function is exerted through the directed secretion of perforin- and granzyme-containing lytic granules following formation of an immunological synapse (IS) that includes adhesion, effector, and termination stages (*Orange, 2008*). Cell adhesion is mediated by integrins, particularly LFA-1, that play a critical role in IS formation and actin polymerization. Prior to the polarization of the microtubule organizing center (MTOC) to the IS, lytic granules converge to the MTOC in a dynein-dependent minus-ended directed manner that is independent of actin polymerization and microtubule dynamics; as such, ligation of LFA-1 or CD28 alone is sufficient to induce convergence (*James et al., 2013*; *Mentlik et al., 2010*). LFA-1-mediated outside-in signaling leads to F-actin polymerization and reorganization, phosphatidylinositol 4,5-bisphosphate (PtdIns(4,5)P2) generation, and the activation of protein tyrosine kinases, including Src family kinases (*Mace et al., 2009*; *Steblyanko et al., 2015*). Activating signaling and sustained F-actin remodeling promotes the polarization of the MTOC, with converged lytic granules, to the lytic IS (*Chen et al., 2007*; *Krzewski et al., 2008*; *Lagrue et al., 2013*). Ultimately, lytic granules transverse a dynamic yet pervasive cortical actin network to access the plasma membrane, where they undergo SNARE-mediated exocytosis at the IS (*Brown et al., 2011*; *Carisey et al., 2018*; *Krzewski and Coligan, 2012*; *Rak et al., 2011*).

Previous studies have demonstrated that CD56 can promote human NK cell cytotoxic function against some CD56$^+$ target cells (*Jarahian et al., 2007*; *Nitta et al., 1989*; *Taouk et al., 2019*; *Valgardsdottir et al., 2014*), although in some cases lysis is not enhanced by target cell CD56 expression (*Lanier et al., 1991*). As these studies focused on homotypic CD56-mediated interactions, the significance of CD56 binding to heterotypic ligands is unclear yet is likely relevant. In neuronal cells, CD56 binds FGFR1 in cis and in trans, and CD56 on NK cells binding to FGFR1 on T cells leads to IL-2 production (*Kos and Chin, 2002*). CD56 also mediates direct recognition of *A. fumigatus* leading to fungal-induced NK cell production of MIP-1α, MIP-1β and RANTES (*Ziegler et al., 2017*). This interaction is marked by accumulation of CD56 at the interface between the NK cell and *A. fumigatus* and is actin-dependent (*Ziegler et al., 2017*).

While CD56 has been implicated in NK cell development, migration, and cytotoxicity (*Nitta et al., 1989*; *Taouk et al., 2019*; *Lanier et al., 1991*; *Chen et al., 2018*; *Mace et al., 2016*), the signaling pathways that regulate its function in immune cells have not been described. Given signaling downstream of CD56 that is mediated by FAK in neuronal cells, one potential link between CD56 and IS formation is the closely related non-receptor tyrosine kinase 2 (Pyk2), which is highly expressed in NK cells (*Gismondi et al., 1997*). FAK and Pyk2, with its expression more restricted to cells of hematopoietic origin, play critical roles in cell adhesion, cell migration and actin remodeling. Stimulation or engagement through multiple receptors, including T cell receptors, integrins and G protein coupled receptors, leads to Pyk2 phosphorylation and activation. As has been reported for Fyn-dependent activation of FAK in neuronal cells, tyrosine 402 (Y402) of Pyk2 is a substrate for Fyn-dependent signaling downstream of TCR ligation (*Qian et al., 1997*). In addition, Pyk2 clustering leads to rapid autophosphorylation on Y402 by trans-acting intermolecular interactions (*Eide et al., 1995*; *Park et al., 2004*). Phosphorylation on Pyk2 Y402, which is equivalent to Y397 of FAK, enables binding and activation of SH2 domain-containing proteins, including Src kinases, and downstream activation of multiple signaling pathways that mediate cell adhesion and migration (*Parsons, 2003*). In NK cells, Pyk2 is phosphorylated downstream of integrin β2 ligation as part of an ILK-PINCH-PAR-VIN signaling cascade that leads to activation of Cdc42, which can control microtubule dependent polarity through CLIP-170 and actin remodeling through WASp and the Arp2/3 complex (*Zhang et al., 2014*). Pyk2 colocalizes with the MTOC in the uropod of migrating NK cells, however following activation it is translocated to the IS and is required for MTOC polarization in IL-2 activated primary NK cells (*Sancho et al., 2000*). Expression of dominant negative Pyk2 disrupts cytotoxicity in this system, and its interactions with β1 integrin, paxillin, and other protein tyrosine kinases suggests that Pyk2 plays a role as a scaffolding protein that helps orchestrate NK cell cytotoxicity (*Gismondi et al., 1997*; *Zhang et al., 2014*; *Sancho et al., 2000*).

Here, we describe a requirement for CD56 in human NK cell function and show that deletion of CD56 in two human NK cell lines leads to impaired secretion and accompanying lytic dysfunction. Furthermore, we identify Pyk2 as a critical signaling intermediate downstream of CD56. These data demonstrate a direct role for CD56 in the NK cell-mediated lysis of CD56-negative target cells and describe a novel activation pathway for cytotoxicity that is unique to human NK cells.

## Results

### Characterization of CD56 expression and polysialation in primary cells and NK cell lines

We previously used CRISPR-Cas9 to generate stable CD56-knockout (KO) NK92 cell lines and define a requirement for CD56 in human NK cell migration (*Mace et al., 2016*). To extend our findings to a second NK cell line, we generated YTS CD56-KO cell lines using the same approach and CRISPR guides. CD56-negative YTS cells were isolated by FACS and the absence of CD56 protein was confirmed in both YTS and NK92 CD56-KO cell lines by Western blot analysis and flow cytometry (*Figure 1A,B*).

While transcripts for all three commonly expressed NCAM isoforms (NCAM140, NCAM180, NCAM120) can be detected in human NK cells, NCAM140 has been previously reported to be the only isoform expressed (*Lanier et al., 1989*; *Lanier et al., 1991*). The extracellular domain of NCAM can be also post-translationally modified by the addition of polysialic acid (PSA), which affects the molecular weight of CD56 when detected by Western blotting. We noted that Western blot analyses of NK92 and YTS cell lines suggested that CD56 is highly polysialated, particularly in the NK92 cell line, leading to a range of apparent molecular weights including a band corresponding to the 140 kDa isoform (*Figure 1A*). In contrast, the YTS cell line expressed primarily the 140 kDa isoform with less polysialation, whereas primary NK cells expressed higher molecular weight isoforms and, as previously reported, also expressed polysialated CD56 (*Figure 1A*; *Moebius et al., 2007*). To determine the contribution of polysialation to the molecular weight of CD56 on YTS and NK92 cell lines, we performed enzymatic treatment of cell lysates with PNGase F to cleave polysialic acid followed by Western blotting with anti-CD56 antibody. These data showed that removal of PSA reduced the variability of CD56 molecular weights and suggested that the 140 kDa or 120 kDa isoform were predominantly expressed in NK cell lines (*Figure 1C*). Similar treatment followed by immunoblotting with a PSA-NCAM-specific antibody demonstrated that PSA-NCAM was not detectable following PNGase F treatment (*Figure 1—figure supplement 1*), thus confirming that the treatment was highly effective in PSA removal.

Despite the reduction in molecular weight variability generated by enzyme treatment, it was difficult to define whether the most predominant band in the cell lines was the 120 kDa or 140 kDa isoform. To determine whether NCAM120, which is GPI-anchored, was expressed on the cell surface of NK cell lines, we treated cells with phosphoinositide phospholipase C (PI-PLC) and quantified expression of NCAM on the cell surface by flow cytometry. CD56 expression on WT NK92 and YTS cells was resistant to PI-PLC cleavage while CD55, a GPI-anchored protein, present on Raji and Jurkat cells, was cleaved (*Figure 1D*). Together, these data strongly suggest that the isoform expressed on the surface of NK92 and YTS cells is not GPI-anchored NCAM120, but is primarily NCAM140 as previously reported (*Kleene and Schachner, 2004*; *Lanier et al., 1991*).

### CD56-KO NK cells have impaired lytic function towards CD56-negative targets

Previous reports on the role of CD56 in NK cell lytic function have focused on homotypic interactions between CD56 on NK cells and target cells and have led to conflicting conclusions about the role that CD56 plays in cytotoxicity (*Nitta et al., 1989*; *Taouk et al., 2019*; *Valgardsdottir et al., 2014*; *Lanier et al., 1991*). To test the cytotoxic function of wild-type (WT) and CD56-knockout (KO) cell lines against CD56-negative target cells, we performed $^{51}$Cr-release cytotoxicity assays. NK92 or YTS cells were used as effectors against K562 target cells, which are susceptible to NK92-mediated lysis, or 721.221, which are susceptible to lysis by both cell lines (*Gwalani and Orange, 2018*). Deletion of CD56 in the NK92 cell line severely abrogated its cytolytic function against susceptible K562 and 721.221 target cell lines (*Figure 2A*). However, YTS CD56-KO cells exerted normal lytic function

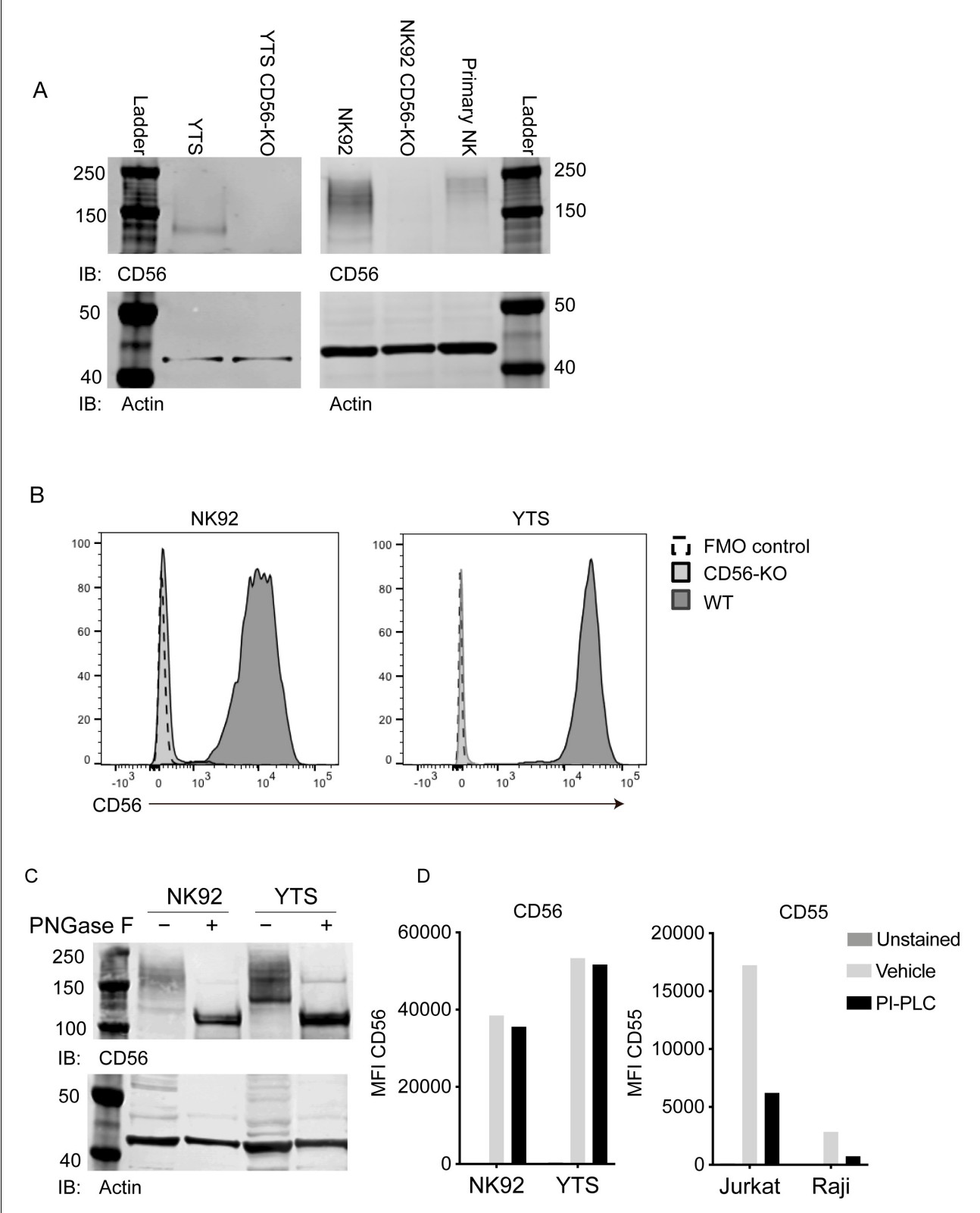

**Figure 1.** Validation of CD56 deletion in human NK cell lines and characterization of CD56 and its polysialation in human NK cells. (**A**) Western blot analysis of CD56 from wild-type (WT) and CD56-knockout (KO) YTS (left) and NK92 (right) cell lines or primary human NK cells with actin as a loading control. (**B**) Flow cytometry analysis of CD56 expression in NK92 or YTS WT (filled histogram, dark grey) or CD56-KO (filled histogram, light grey) cells compared to unstained cells (dashed line). (**C**) NK92 or YTS cells were treated with PNGase F to remove polysialic acid. Following treatment, lysates

*Figure 1 continued on next page*

*Figure 1 continued*

were separated by SDS-PAGE and CD56 or actin as a loading control were detected by Western blotting. (**D**) NK cell lines (left) or Jurkat or Raji cells as a positive control (right) were treated with PI-PLC to cleave GPI anchored proteins from the cell surface. PI-PLC activity was confirmed by cleavage of GPI-anchored CD55 (right). All data shown are representative of 3 technical replicates performed on different days.

The online version of this article includes the following figure supplement(s) for figure 1:

**Figure supplement 1.** Expression of PSA-NCAM on human NK cell lines.

against 721.221 target cells when compared to their WT counterparts (*Figure 2B*). To test a second susceptible target with YTS cells, we used KT86 cells, which are K562 cells that express CD86 and thus are susceptible to YTS-mediated lysis (*Banerjee et al., 2007*). The use of KT86 targets did not reveal a requirement for CD56 in their lysis by YTS cells despite the failure of NK92 CD56-KO cells to lyse K562 targets (*Figure 2B*). These data demonstrated the conserved defect in NK92 CD56-KO cytolytic function, whereas YTS cells were not significantly affected by loss of CD56 in their killing of multiple target cell lines in a 4 hr $^{51}$Cr assay.

NK cell cytotoxic function includes serial killing of target cells, with target cell lysis followed by NK cell detachment and re-engagement with subsequent targets (*Anft et al., 2020*; *Bhat and Watzl, 2007*; *Jenkins et al., 2015*; *Vanherberghen et al., 2013*). Given the demonstrated defect in NK cell migration in CD56-KO NK92 cells (*Mace et al., 2016*) we sought to measure whether the defect in NK cell killing could be attributed to impaired serial killing. The average time to first kill of a target cell by an NK92 or YTS cell ranges from 35 to 45 min (*Gwalani and Orange, 2018*), therefore to test the efficacy of early cytotoxic function we performed $^{51}$Cr cytotoxicity assays with 1 hr incubations with target cells. As seen in 4 hr assays, the lytic function of NK92 CD56-KO cells was severely impaired following 1 hr of incubation with target cells, suggesting that the decrease seen at 4 hr is not due to defects in serial killing. In addition, 1 hr $^{51}$Cr-release assays with YTS CD56-KO cells revealed a decreased specific lysis compared to WT YTS (*Figure 2C*). While the killing decrease in the YTS CD56-KO cells at 1 hr was not as substantive as observed in NK92 CD56-KO cells, it was statistically significant at all effector to target ratios and consistent between multiple experiments. Therefore, while the impairment in CD56-deficient NK92 cells was more profound, there was a conserved phenotype between the two human NK cell lines tested.

To determine whether the observed defect in cytotoxicity was due to dysregulated expression of molecules associated with NK cell lytic function, multiparametric flow cytometry was performed using panels designed to measure expression of receptors required for NK cell development, adhesion, activation, and inhibition (*Mahapatra et al., 2017*). Small increases in the percent positive cells and mean fluorescence intensity of CD2, CD11a, CD18, CD45, CD94, and NKG2A were observed in NK92 CD56-KO cells (*Figure 2—figure supplement 1*). However, these slight differences may be attributed to better ligand accessibility by antibodies, as deleting CD56 removes the long chains of negatively charged polysialic acid attached to CD56 (*Nicoll et al., 2003*). No significant differences were observed in the frequency of NK92 CD56-KO cells that were positive for expression of granzyme A or B and for IFNγ at rest or after activation with PMA/ionomycin relative to the parental NK92 cell line ((*Figure 2—figure supplement 1*). Similarly, we did not observe differences in surface receptors or effector molecules at rest or after activation in YTS CD56-KO cells when compared to wild-type YTS ((*Figure 2—figure supplement 1*). Therefore, we concluded that deletion of CD56 did not lead to dysregulated expression of several common NK cell receptors and, importantly, the capacity to produce granzymes A and B and interferon gamma was retained in CD56-KO cells.

To confirm that the defect in NK92 cytotoxicity was specifically conferred by CD56 deletion, NK92 CD56-KO cells were retrovirally transduced with the NCAM140 isoform fused with an mApple fluorescent reporter to re-introduce CD56 expression. CD56 protein expression and expression of polysialic acid were confirmed in the reconstituted cells by flow cytometry ((*Figure 2—figure supplement 2*). Importantly, re-expression of CD56 restored the lytic function of NK92 CD56-KO cells against both 721.221 and K562 targets (*Figure 2D*), demonstrating that the impairment we observed was specific to the deletion of CD56 and that re-expression of NCAM140 was sufficient to rescue this defect.

We sought to further define the requirements for the intracellular and extracellular domains of CD56 in NK cell cytotoxicity. The 140 kDa isoform of CD56 (NCAM) expresses 5 Ig-like domains and

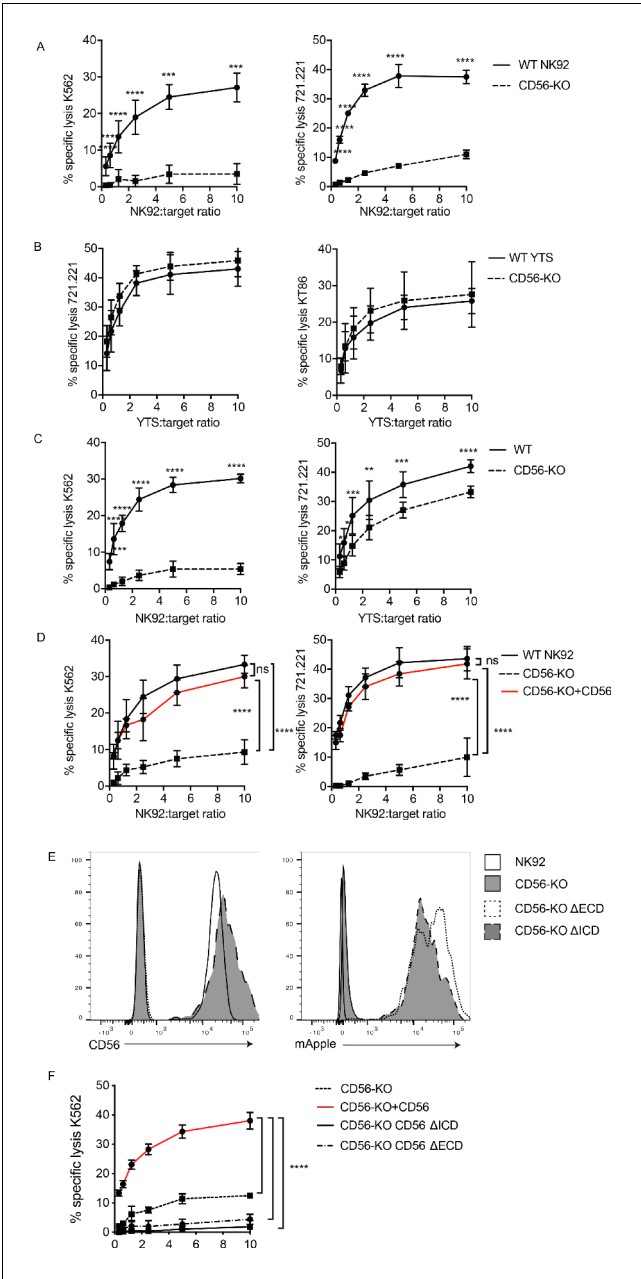

**Figure 2.** CD56 deletion abrogates NK92 cytotoxic function and delays YTS cytotoxicity. $^{51}$Cr-release assays were performed using NK92 (**A, C, D**) or YTS (**B, C**) WT and CD56-KO cell lines as effectors against susceptible targets. (**A**) 4 hr assays were performed with NK92 cell lines using K562 (left) or 721.221 (right) target cells. (**B**) 4 hr assays were performed with YTS cell lines using 721.221 (left) or KT86 (right) target cells. (**C**) 1 hr $^{51}$Cr-release assays were performed using NK92 (left) or YTS (right) cells as effectors. (**D**) CD56 (NCAM-140) was re-expressed in NK92 CD56-KO cells and these cells, NK92 or NK92 CD56-KO cells were used for 4 hr cytotoxicity assays against K562 (left) or 721.221 (right) target cells. (**E**) CD56-KO NK92 cells were transfected with chimeric CD56 constructs fused to an mApple fluorescent reporter as described in Materials and methods. Flow cytometry was used to confirm the expression of CD56 and/or mApple. (**F**) Cytotoxicity assays were performed with chimeric cell lines using K562 cells as targets. Mean ± S.D. of three independent experiments pooled. **p<0.01, ***p<0.001, ****p<0.0001 by Ordinary one-way ANOVA with multiple corrections test or unpaired student t-test with Welch's correction. ΔECD: chimeric construct lacking extracellular domain, ΔICD: chimeric construct lacking intracellular domain.

The online version of this article includes the following figure supplement(s) for figure 2:

**Figure supplement 1.** Phenotyping of NK cell lines by flow cytometry.

**Figure supplement 2.** Re-expression of CD56 and PSA-NCAM in reconstituted NK92 cell lines.

2 FNIII repeats, a transmembrane domain and an intracellular domain. We generated CD56 chimeric constructs and retrovirally transduced these into NK92 CD56-KO cells. ΔICD lacks the intracellular domain (ICD) of CD56 but contains the extracellular domain (ECD) and transmembrane domain (TM); ΔECD lacks the CD56 ECD but includes the transmembrane and intracellular domains. Stable NK92 CD56-KO cell lines were generated and confirmed to be mApple positive and the ΔICD cells were verified to be CD56 positive (*Figure 2E*). $^{51}$Cr-release assays were performed to determine the requirements for CD56 domains in cytotoxicity in NK92 cells. Sole expression of either the ECD or ICD was not sufficient to rescue cytotoxicity (*Figure 2F*), whereas full-length CD56 restored cytotoxicity as previously demonstrated (*Figure 2D,F*). Collectively, these data indicate that the full-length NCAM140 protein is required to promote human NK cell cytotoxicity and suggests a role for intracellular interactions that mediate this process.

## Secretory function is decreased in NK92 and YTS CD56-KO cell lines

Both NK92 and YTS cell lines kill target cells via perforin- and granzyme-dependent exocytosis. In addition, YTS in particular are potent producers of IFNγ upon stimulation (*Gunesch et al., 2019*). To further assess the effect of CD56 deletion on NK cell function, two methods of detection were used; a colorimetric benzyloxycarbonyl-L-lysine thiobenzyl ester (BLT) esterase assay for detection of granzyme A function as a readout for granule exocytosis and detection of surface-exposed lysosomal-associated membrane protein-1 (LAMP-1 or CD107a) by flow cytometry after plate-bound activation or coculture with target cells (*Alter et al., 2004*; *Betts and Koup, 2004*; *Suhrbier et al., 1991*). We activated WT, CD56-KO, and CD56 rescued CD56-KO NK92 cells for 90 min with immobilized anti-CD18 and -NKp30 antibodies and measured BLT esterase activity (*Rak et al., 2011*). As suggested by the impaired lytic function demonstrated by $^{51}$Cr assays, NK92 CD56-KO cells had significantly decreased release of BLT esterase compared to WT NK92, demonstrating that degranulation is impaired in the CD56-KO cells. Re-expression of full-length CD56 in NK92 CD56-KO cells restored esterase activity comparable to that of WT NK92 (*Figure 3A*).

To further support our observation that CD56-KO NK92 cells did not undergo lytic granule exocytosis, we measured CD107a exposure following activation by plate-bound antibodies or target cells. Following plate-bound stimulation, NK92 CD56-KO had a significantly reduced percentage of cells positive for CD107a compared to WT NK92, and those cells that did express CD107a did so with a lower mean fluorescent intensity, supporting our data showing decreased secretion in the absence of CD56 (*Figure 3B*). As with the BLT esterase assay, re-expression of CD56 in the knockout cells restored their capacity to degranulate (*Figure 3B*). Furthermore, the same effect was observed when NK92 cells were co-incubated with target cells (*Figure 3C*). Taken together, these data demonstrated that the deletion of CD56 in NK92 cells leads to a defect in lytic granule exocytosis in response to target cell activation or activating receptor cross-linking.

We failed to detect BLT esterase secreted from YTS cell lines, likely due to their very low expression of granzyme A (*Gunesch et al., 2019*). Furthermore, YTS cells undergo fewer degranulation events on a per cell basis when killing target cells than NK92 (*Gwalani and Orange, 2018*). Measurement of exposed CD107a on YTS cell lines (WT and CD56-KO) after plate-bound activation proved difficult and inconclusive after coculture with 721.221 target cells (data not shown). However, YTS cells robustly produce IFNγ in response to activation (*Gunesch et al., 2019*). Co-incubation of WT YTS with 721.221 target cells led to secretion of IFNγ detectable by ELISA, whereas this secretion was significantly decreased in the CD56-KO YTS cells (*Figure 3D*). We confirmed robust IFNγ production in YTS CD56-KO cells in response to stimulation with PMA/ionomycin by intracellular flow cytometry ((*Figure 2—figure supplement 1*), confirming that this decrease in IFNy levels from YTS cells conjugated with 721.221 is due to a defect in secretion, rather than production. Therefore, despite their intact target cell killing in 4 hr $^{51}$Cr assays, YTS cells had significant impairment in secretion in response to contact-dependent activation when cytokine production was measured.

In summary, deletion of CD56 in the NK92 and YTS cell lines impairs their secretion in response to activation, including degranulation in NK92 CD56-KO cells, demonstrating a critical role for CD56 in NK function that is independent from CD56 homotypic interactions. The differential functional response to CD56 deletion of these cell lines supports their differing properties and speaks to relevant differences in their biology (*Gunesch et al., 2019*); here we have chosen to focus on the mechanism by which CD56 is mediating lytic function in the NK92 cell line.

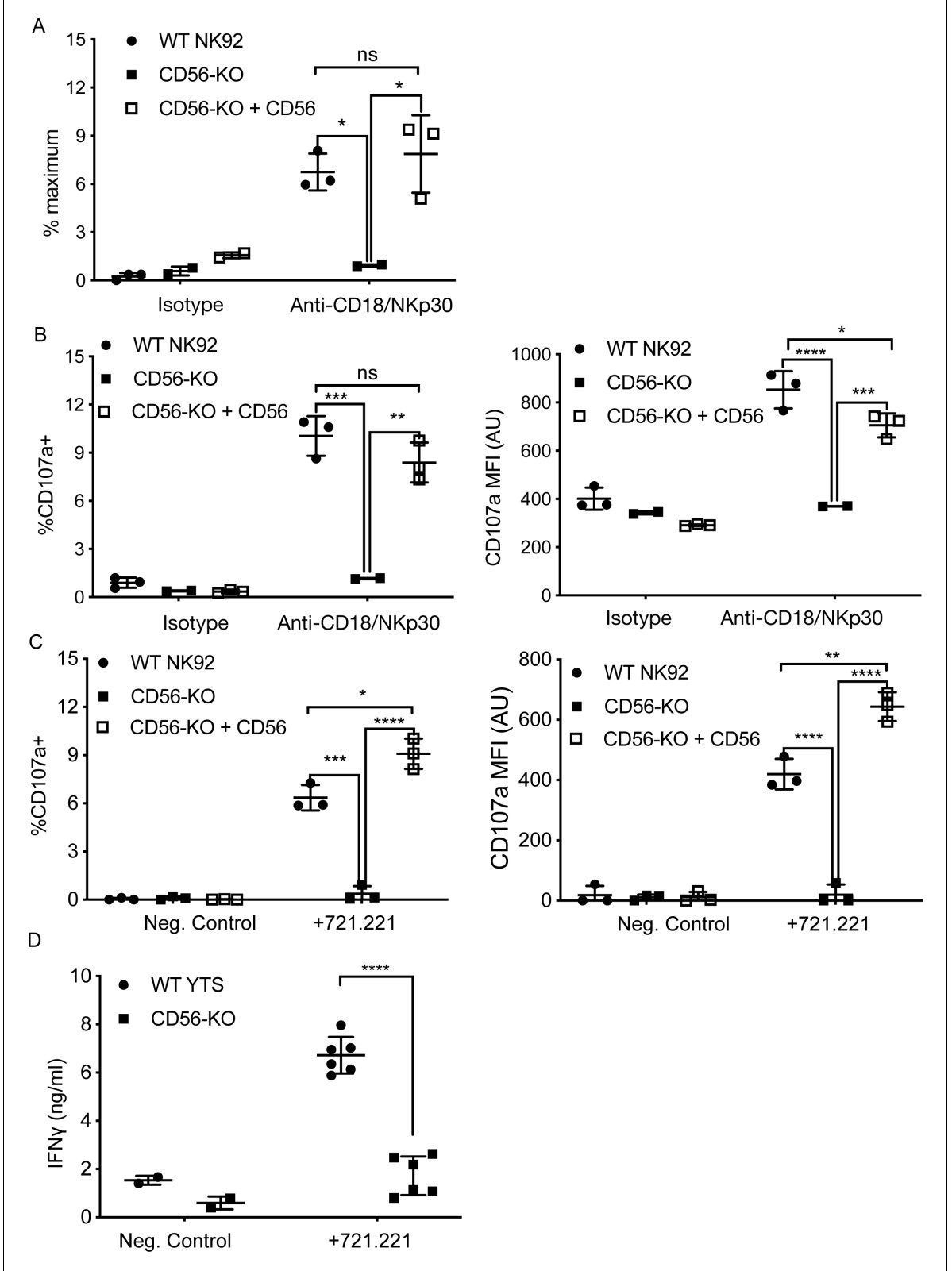

**Figure 3.** CD56 expression is required for exocytosis of NK92 cells. (**A**) WT NK92, CD56-KO or CD56-KO reconstituted cells were incubated for 60–90 min on plates pre-coated with 10 µg/ml of anti-CD18 and anti-NKp30 antibodies. Supernatant was collected and granzyme A secretion was measured by a BLT esterase assay. Secretory potential was measured as a readout of the % maximum of granzyme A activity in the supernatant. (**B**) WT NK92, CD56-KO or CD56-KO reconstituted cells were incubated for 1–2 hr on plates pre-coated with 10 µg/ml of anti-CD18 and anti-NKp30 antibodies. Cells

*Figure 3 continued on next page*

*Figure 3 continued*

were harvested and degranulation was measured by CD107a expression using flow cytometry. (**C**) WT NK92, CD56-KO or CD56-KO reconstituted cells were co-cultured with 721.221 target cells. Cells were harvested and CD107a expression was measured by flow cytometry. For co-culture experiments the average background of media only was subtracted from samples. Mean ± SD. *p<0.05, **p<0.01, ***p<0.001, ****p<0.0001 by one-way ANOVA with Tukey's multiple comparisons post-hoc test. (**D**) WT YTS and CD56-KO cells were incubated with 721.221 target cells at a 2:1 ratio for 22 hr. Supernatant was collected and used in a human IFN gamma ELISA. ****p<0.0001 by unpaired student t-test with Welch's correction. All data are representative of 3 independent experiments performed in duplicate or triplicate.

## Lytic immune synapse formation and function

NK cell lytic function is exerted through the formation of an immunological synapse, which serves to focus directed secretion and mitigate bystander killing. The steps leading to IS formation can broadly be defined by adhesion/activation, polarization, secretion, and termination stages (*Orange, 2008*). Initial activation through ITAM-based activating receptors, integrins, or cytokine stimulation leads to lytic granule convergence to the MTOC, a step that precedes actin remodeling and the re-orientation of the MTOC to the synapse and ultimately lytic granule exocytosis and target cell death.

Having shown that the secretion stage was impacted by CD56 deletion, we sought to further define how activation and polarization were affected in NK92 CD56-KO NK cell lines. Fixed cell confocal microscopy was performed to visualize granule convergence, MTOC polarization, and actin accumulation at the cell-cell interface and each of these parameters was quantified as previously described (*Hsu et al., 2017*; *Sanborn et al., 2010*; *Banerjee and Orange, 2010*). Both WT and CD56-KO NK92 cells were found in conjugates with K562 target cells, suggesting that their ability to adhere to targets was not impaired (*Figure 4A*). While actin remodeling was consistently decreased in CD56-KO cells relative to WT NK92, this effect was not significant (*Figure 4B*). In contrast, the MTOC was not polarized towards the immunological synapse in CD56-KO NK92 cells as it was in NK92 cells, suggesting decreased activation. This observation was quantified by significantly greater MTOC-synapse distance (*Figure 4C*). However, convergence of lytic granules to the MTOC was not affected by deletion of CD56 (*Figure 4D*).

Given the importance of cell adhesion in IS formation, we sought to further confirm that impaired conjugate formation between CD56-KO NK92 cells and their targets was not underlying their observed cytotoxic defect. We performed flow cytometry-based conjugate assays in which effector and target cells were differentially labeled, co-incubated, and then fixed prior to flow cytometric analyses. These analyses demonstrated that there was not an impairment in conjugate formation and that the frequency of conjugates formed with CD56-KO cells was, in fact, higher at 30- and 60 min timepoints but not significantly different at 10- and 120 min timepoints (*Figure 4E*). Therefore, loss of CD56 expression in the NK92 cell line, which results in impaired cytotoxic function, is accompanied by impaired formation of the immunological synapse reflected by reduced MTOC polarization towards target cells despite intact conjugate formation.

## Pyk2 colocalizes with CD56 and its phosphorylation is decreased in NK92 CD56-KO cells

The introduction of the CD56-domain specific constructs into the NK92 CD56-KO cells illustrated the observation that both the extracellular and intracellular domain of CD56 are required to recover cytotoxicity. This suggests that the intracellular domain of CD56 may be interacting with intracellular molecules within NK92 cells to enable cytotoxic function. Pyk2 is closely related to FAK and a predicted interacting partner of CD56 through binding to Fyn (*Sancho et al., 2000*; *Szklarczyk et al., 2015*). In non-immune cells, engagement of NCAM140 (CD56) recruits and activates FAK, which in turn activates the MAPK signaling pathway that is important for neurite outgrowth and cell survival (*Beggs et al., 1997*; *Schmid et al., 1999*; *Ditlevsen et al., 2003*). Additionally, Pyk2 is an important regulator of NK cell cytotoxicity by stimulating lytic granule polarization and target cell conjugation (*Zhang et al., 2014*; *Sancho et al., 2000*; *Gismondi et al., 2000*). To determine the effect of CD56 deletion on Pyk2 phosphorylation WT, CD56-KO, and CD56-KO reconstituted NK92 cells were activated by plate-bound anti-NKp30 and -CD18. Cells were dissociated and phosphorylation of Pyk2 (Y402) was measured by intracellular flow cytometry. We found that phosphorylation of Pyk2 Y402

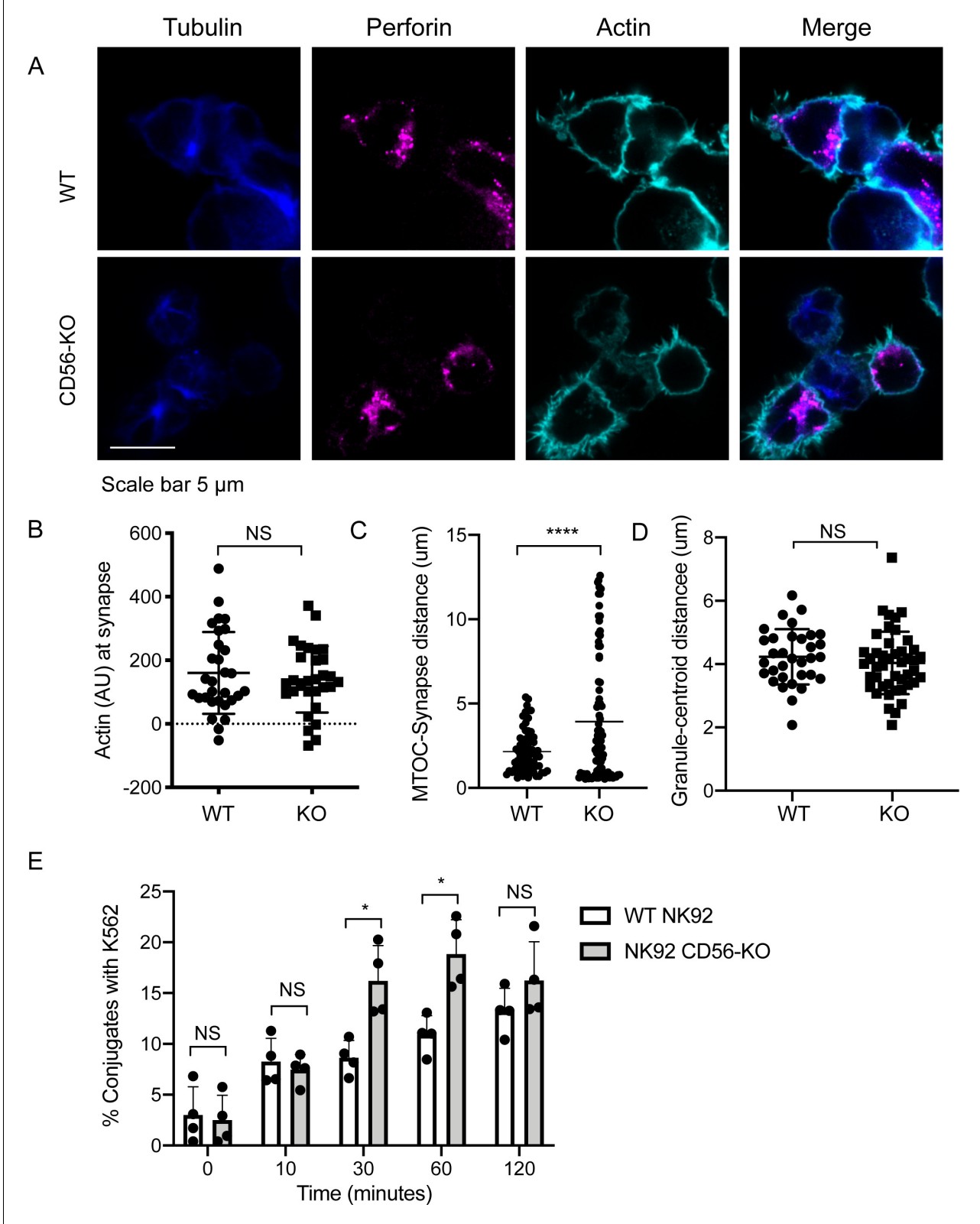

**Figure 4.** Impaired immune synapse formation in NK92 CD56-KO cells. WT or CD56-KO effector cells were cultured at a 2:1 ratio with K562 target cells for 45 min then fixed, immunostained and visualized by confocal microscopy. (**A**) Representative images from >30 cells from three independent experiments immunostained as indicated. (**B**) Integrated intensity of actin at the immune synapse for WT or CD56-KO cells. Data are representative from one experiment performed three times. n = 30, 39. NS = not significant by unpaired t-test. (**C**) MTOC to synapse distance (μm)
*Figure 4 continued on next page*

*Figure 4 continued*

calculated from WT or CD56-KO conjugates. n = 70, 83. Data pooled from three independently replicated experiments. (**D**) Mean granule to centroid distance for WT or CD56-KO conjugates. Each data point represents the mean distance granule to centroid distance from one conjugate. n = 33, 45 from one representative experiment of >3 experiments. NS = not significant by unpaired T test with Welch's correction. (**E**) WT or CD56-KO NK92 effector cells were differentially labeled then conjugated at a 2:1 ratio with K562 target cells for the times indicated then fixed and analyzed by flow cytometry. The frequency of NK92-K562 conjugates was calculated for each timepoint. Each point represents a single experiment performed on different days in triplicate (n = 4 replicates). Error bars indicate mean ± SD; *p<0.05 by Mann-Whitney test.

was significantly decreased in the NK92 CD56-KO cells, a phenotype that was recovered by the reconstitution of CD56 (*Figure 5A*).

We hypothesized that the differential impact upon NK cell function on YTS and NK92 cell lines following CD56 deletion may be related to differential usage of Pyk2-dependent signaling pathways. To test this hypothesis, we analyzed Pyk2 phosphorylation in YTS and NK92 WT and CD56-KO cells using flow cytometry as described above. While we noted a similar trend in YTS cells, notably decreased Pyk2 phosphorylation in CD56-KO cells relative to WT, this effect was not as profound as that seen in NK92 cells and was not statistically significant (*Figure 5B*). In addition, the magnitude of Pyk2 phosphorylation in WT YTS cells was lower than that of NK92 cells based upon normalized fluorescence intensities. The lower relative fluorescence intensity of Pyk2 Y402 phosphorylation in YTS cells and greater relative impairment in NK92 CD56-KO cells suggests a greater dependence on Pyk2 phosphorylation in NK92 cells and a potential mechanism whereby CD56 deletion has a greater effect on Pyk2-mediated cytotoxic function in NK92 than YTS cells. To further investigate this hypothesis, we performed $^{51}$Cr assays in the presence of PF431396, a Pyk2/FAK inhibitor that inhibits Pyk2 autophosphorylation and blocks its kinase function (*Buckbinder et al., 2007*). Consistent with the differential phosphorylation of CD56 in YTS and NK92 cells, the presence of Pyk2 inhibitor significantly decreased the cytotoxic function of WT NK92 cells, whereas both WT and CD56-KO YTS cells were only minimally affected (*Figure 5C,D*). These data demonstrate that, in the absence of CD56 expression in NK92 cells, Pyk2 phosphorylation is decreased and that inhibition of Pyk2 phosphorylation in NK92 cells impairs cellular cytotoxicity. In YTS cells, a reduced dependence on Pyk2 for target cell lysis is reflected by retained lytic function both in the presence of Pyk2 inhibitor and the absence of CD56-mediated function.

We further sought to determine whether Pyk2 localization at the immune synapse was affected by loss of CD56. Fixed cell confocal microscopy was performed by acquiring 3D volumes of WT or CD56-KO NK92 cells conjugated to K562 target cells and immunostained for pPyk2 Y402, perforin and actin. As suggested by intracellular flow cytometric analysis of Pyk2 phosphorylation, we detected higher fluorescence intensity of pPyk2 Y402 in WT NK92 relative to CD56-KO NK92 (*Figure 5E*). Quantification of conjugates confirmed greater pPyk2 accumulation at the IS of WT NK92 cells when compared to CD56-KO NK92 cells (*Figure 5F*). In contrast, we measured no difference in the localization or intensity of total Pyk2 between WT and CD56-KO cell lines, which primarily localized to the MTOC as previously reported (*Sancho et al., 2000*) ((*Figure 5—figure supplement 1*). These measurements were made difficult by the significant phosphorylation of Pyk2 detected in the target cells, however the effect of CD56 deletion on Pyk2 phosphorylation is underscored by our functional experiments in a target cell-free system (*Figure 5A*).

Both Pyk2 and CD56 are localized to the uropod of migrating human NK cells (*Mace et al., 2016*; *Sancho et al., 2000*). We noted that CD56 in NK92 cells conjugated to targets remained partially localized to the uropod, however a fraction of CD56 localized to the IS, where it strongly co-localized with actin and pPyk2 Y402 (*Figure 5E*). This was observed both early and late in immune synapse formation (data not shown), suggesting that the effect we observed was not due to the kinetics of CD56 re-distribution. These data support our functional studies demonstrating that loss of CD56 in NK92 cells impairs immune synapse function through deregulated Pyk2 activation, while additionally illustrating the recruitment of CD56 to the IS in human NK cells.

## Cytotoxic function of CD56-deficient primary NK cells is intact

We sought to determine the effect of CD56 deletion on primary NK cell function. Using CRISPR-Cas9, we deleted CD56 in bulk peripheral blood NK cells or CD56$^{bright}$ NK cells activated with IL-15. CD56 expression was reduced by 9-fold when cells were incubated with low dose IL-15 but was still

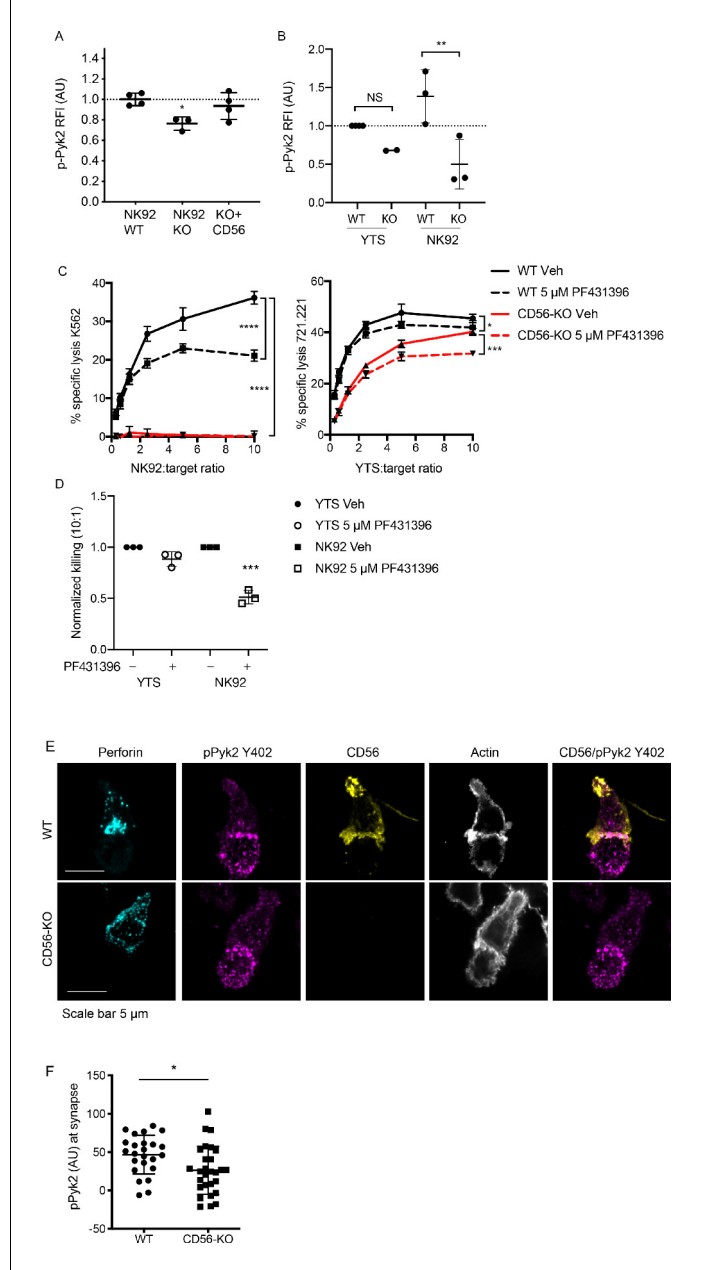

**Figure 5.** Phosphorylation of Pyk2 is decreased in NK92 CD56-KO cells upon activating receptor ligation. (**A**) NK92 WT, CD56-KO or CD56 reconstituted (KO+CD56) cells were incubated for 25–30 min on plates pre-coated with 10 µg/ml of anti-CD18 and anti-NKp30 antibodies. Cells were permeabilized and immunostained for pPyk2 Y402 then data were acquired by flow cytometry. Relative fluorescent intensity (RFI) of pPyk2 was calculated based upon the intensity of the WT NK92 condition. Shown are the pooled data from three independent experiments. (**B**) WT or CD56-KO NK92 or YTS cells were permeabilized and immunostained for pPyk2 Y402 then data were acquired by flow cytometry. Shown is pooled data from 2 (YTS) or 3 (NK92) independent experiments. **p<0.01 by one-way ANOVA with multiple comparisons. (**C**) 4 hr $^{51}$Cr assays were performed with WT (black) or CD56-KO (red) NK92 cells as effectors. Assays were performed in the presence of Pyk2 inhibitor PF431396 or vehicle control (DMSO) following brief pre-incubation of effectors with inhibitor. Shown are representative data from three independent repeats. (**D**) Pooled data from the 10:1 effector to target cell ratio of the experiments described in (**C**) normalized to the WT YTS condition without inhibitor. ***p<0.001 by one-way ANOVA with multiple comparisons. (**E**) Representative confocal microscopy images of WT or CD56-KO NK92 effectors conjugated to K562 target cells in the presence of non-blocking CD56 antibody then fixed and immunostained for perforin, pPyk2 Y402 and actin. (**F**)
*Figure 5 continued on next page*

*Figure 5 continued*

Fluorescent intensity of pPYK2 Y402 at the immune synapse of WT or CD56-KO effector cells. n = 24, 28 from one representative experiment of 3 independent repeats.
The online version of this article includes the following figure supplement(s) for figure 5:

**Figure supplement 1.** Detection of total Pyk2 in WT and CD56-KO cell lines.

detectable on the surface of NK cells (data not shown). Because CD56 protein appeared stable, we utilized higher doses of IL-15 to induce NK cell proliferation and CD56 dilution. Using this approach, the deletion of CD56 was highly efficient following delivery and expansion with IL-15 (*Figure 6A*); notably, NK cell cytotoxic function was intact (*Figure 6B*).

NK92 cells are phenotypically and genotypically more aligned with the CD56$^{bright}$ NK cell subset, whereas YTS cells are more similar to the CD56$^{dim}$ subset (*Gunesch et al., 2019*). Therefore, we reasoned that the effect of CD56 deletion may be greater on the CD56$^{bright}$ subset, especially given its significantly higher expression of CD56. As the CD56$^{dim}$ subset represents the majority of NK cells found within peripheral blood and has significantly decreased expression of CD56 relative to the CD56$^{bright}$ subset, we isolated CD56$^{bright}$ NK cells and repeated the deletion of CD56 and measurement of cytotoxic function following IL-15 expansion. While CD56$^{bright}$ cells do not have significant lytic function when freshly isolated, brief cytokine stimulation with IL-15 is sufficient to confer substantive cytotoxic function on this subset (*Wagner et al., 2017*). Following IL-15 expansion, both mock transfected and CD56-KO CD56$^{bright}$ NK cells showed robust cytolytic function against K562 target cells (*Figure 6C*). Therefore, despite the requirement for CD56 in NK92-mediated cytolytic function, loss of CD56 in CD56$^{bright}$ or CD56$^{dim}$ NK cells does not lead to a defect in primary NK cell cytotoxicity.

We hypothesized that the expansion of primary cells in IL-15 during the process of cell editing may be overcoming the requirement for Pyk2 in primary NK cells, particularly given previous reports of IL-15 signaling altering primary NK cell dependence on Pyk2 (*Lee et al., 2010*). Therefore, we expanded primary NK cells for 15 days in IL-15 and tested the effect of Pyk2 inhibition on cytotoxic function. While Pyk2 inhibition moderately decreased lytic function against K562 target cells following IL-15 activation, the effect was greater in cells that had not been stimulated with IL-15, suggesting that long-term culture in IL-15 can reduce dependency of primary NK cells on Pyk2 Y402 phosphorylation for lytic function (*Figure 6E*).

We further hypothesized that if long-term stimulation in IL-15 could overcome Pyk2 dependency in primary cells, it may be possible to rescue the defect in CD56-KO NK92 cells with IL-15 stimulation. We incubated NK92 WT and CD56-KO cells in IL-15 for 5 days to mimic the conditions used to generate CD56-deficient NK cells. While this stimulation increased the lytic function of WT NK92 cells, activation by IL-15 failed to rescue the cytolytic deficiency in NK92 CD56-KO cells (*Figure 6F*). However, NK92 cells stimulated with IL-15 also remained sensitive to Pyk2 inhibition, suggesting that the mechanism used to overcome Pyk2 dependence in primary cells in response to IL-15 was not functional in NK92 cells.

## CD56 co-localizes to the IS with pPyk2 Y402 in freshly isolated primary NK cells

Despite intact cytolytic function in CD56-deficient primary NK cells, we sought to define the localization of CD56 in primary NK cell conjugates and we performed fixed cell confocal imaging of freshly isolated ex vivo NK cells conjugated to K562 target cells. As with NK cell lines, we found that CD56 was localized to the uropod of migrating cells as previously described (*Sancho et al., 2000*). However, we also frequently found redistribution of a pool of CD56 to the immune synapse (*Figure 7A*). This was quantified by measurement of the mean fluorescence intensity of CD56, which demonstrated greater CD56 intensity at the immune synapse than in non-synaptic regions of the NK cell (*Figure 7B*). Furthermore, CD56 and pPyk2 Y402 were spatially co-localized at the immune synapse, and intensity of pPyk2 was similarly greater at the synapse than in non-synaptic regions as previously described (*Figure 7B*; *Sancho et al., 2000*). Therefore, as with NK cell lines, CD56 and Pyk2 colocalize in primary NK cells and are recruited to the immunological synapse.

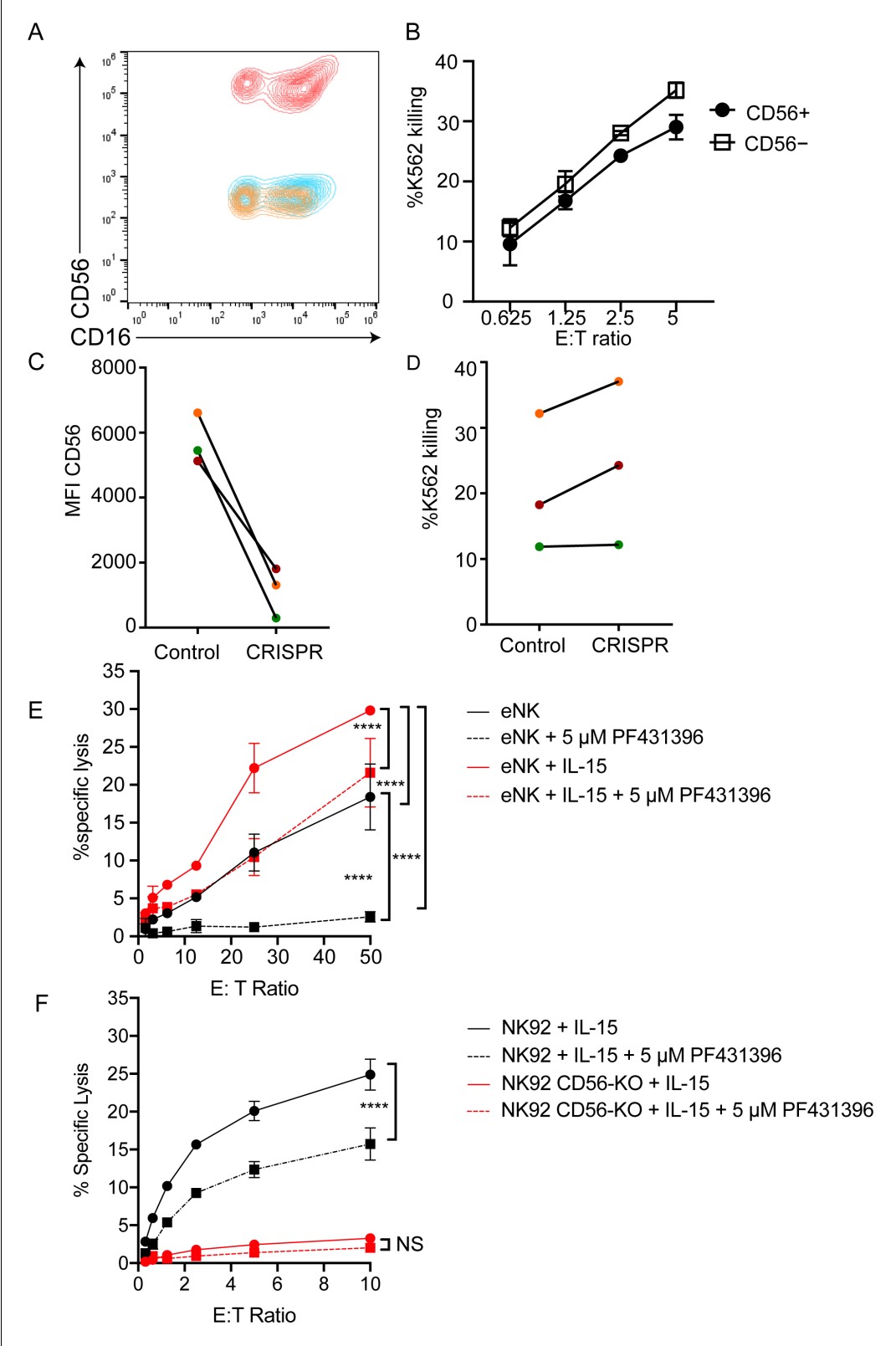

**Figure 6.** CD56-deficient primary NK cells retain lytic function. Primary NK cells were isolated and allowed to rest overnight in the presence of low-dose IL-15 prior to delivery of CD56 CRISPR-Cas9. Cells were further expanded in the presence of 25 ng/ml IL-15 for 15 days and cytotoxicity against K562 targets was measured. (**A**) Representative FACS plot of CD56-deficient (blue) or control primary cells (red) after 15 days of IL-15 expansion. Shown also is the fluorescence minus one control (yellow). (**B**) K562 target cell lysis by primary NK cells shown in (**A**). (**C**) Control or CD56-deficient NK cells from

*Figure 6 continued on next page*

*Figure 6 continued*

three healthy donors were incubated for 1 week after CD56 CRISPR-Cas9 delivery in 25 ng/mL IL-15 then cells were isolated by FACS and cultured for an additional 8 days and the MFI of CD56 was measured by flow cytometry. (**D**) Specific lysis of K562 target cells by isolated and expanded CD56[bright] NK cells from the three healthy donors shown in (**C**). (**E**) Primary NK cells were incubated and expanded for 14 days in the presence of 50 ng/ml IL-15 then cytotoxicity against K562 target cells was tested in the presence or absence of Pyk2 inhibitor PF431396. Freshly isolated, non-expanded NK cells were used as a control and similarly treated with PF431396. (**F**) WT or CD56-KO NK92 cells were incubated for 7 days in the presence of 50 ng/ml IL-15 then cytotoxicity was tested in the presence or absence of PF431396. Shown is one representative experiment from three independent biological repeats. Error bars represent 3 technical repeats, SEM.

## Discussion

While CD56 is the prototypical identifier of human NK cells in peripheral blood, its function has been poorly defined. Early studies of its role in cytotoxicity largely focused on CD56 homotypic interactions and found that cytotoxicity of IL-2 expanded primary NK cells was diminished against CD56[+] target cells in the presence of anti-CD56 blocking antibodies (*Nitta et al., 1989*). In contrast, Lanier et al. found no significant difference in NK cell-mediated lysis of CD56[−] KG1a and CD56[+] KG1a target cells (*Lanier et al., 1991*). In both cases, CD56 functionality was tested with IL-2 expanded

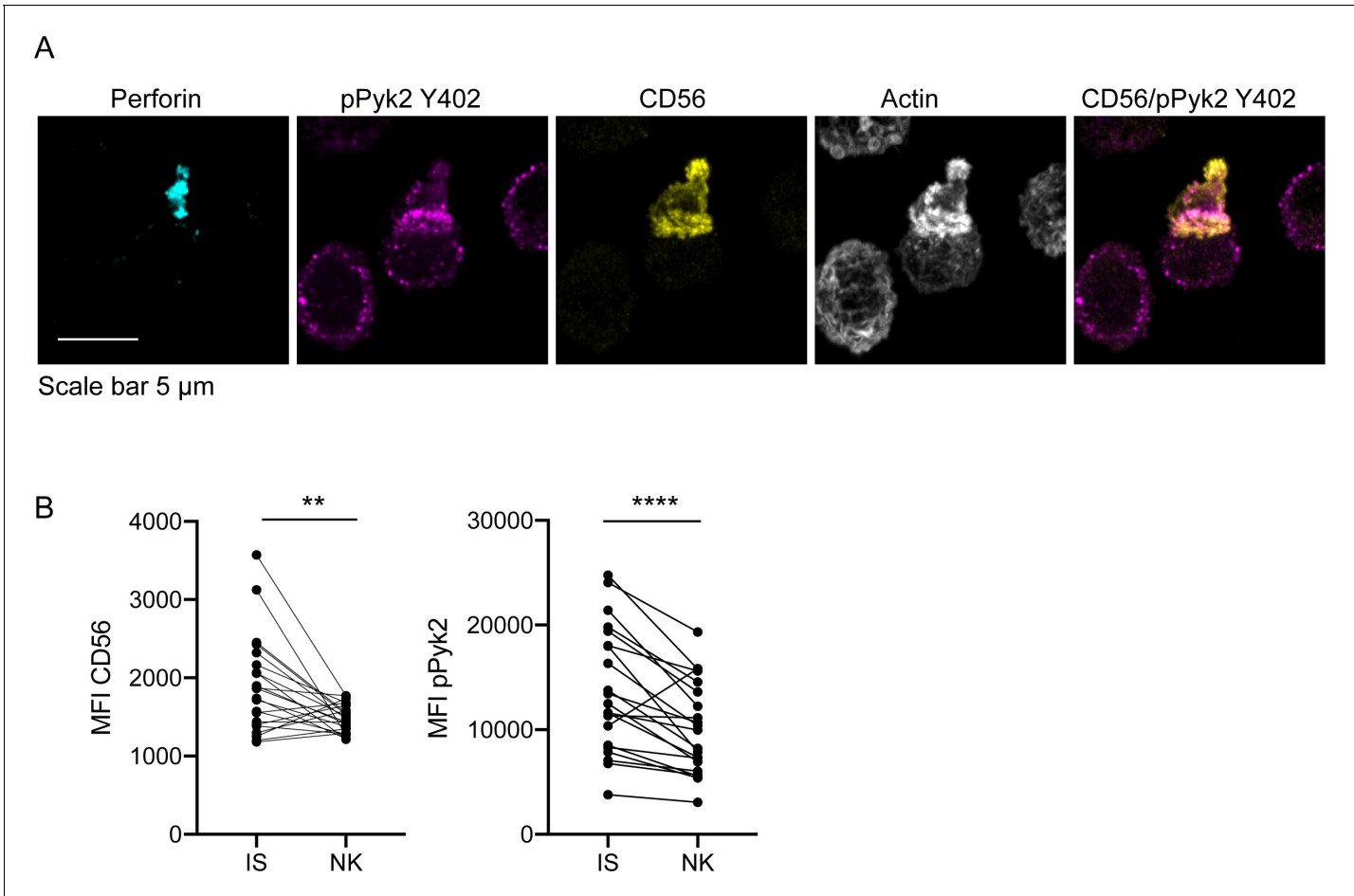

**Figure 7.** CD56 co-localizes with pPyk2 Y402 to the immune synapse in primary human NK cells. Primary NK cells were enriched from peripheral blood then incubated with K562 target cells for 45 min on poly-L-lysine coated coverslips in the presence of non-blocking anti-CD56 antibody. Following incubation cells were fixed, permeabilized, and immunostained for pPyk2 Y402 (magenta), perforin (cyan) and actin (phalloidin, greyscale). 3D volumetric images were acquired by spinning disk confocal microscopy. (**A**) Representative images from one of three healthy donors. Shown is a maximum projection of 13 planes taken with 0.5 µm steps. (**B**) Fluorescence intensity of CD56 (left) or pPyk2 Y402 (right) measured at the synaptic (IS) or non-synaptic (NK) cell cortex of primary NK cells conjugated to target cells. n = 22 from one independent experiment of 3 using three different healthy donors. **p<0.005, ****p<0.0001 by paired t-test.

primary NK cells using antibody to inhibit CD56 homotypic interactions, through the manipulation of CD56 expression on target cells, or a combination of both. CD56 was also implicated in alloantigen-specific recognition by human NK cells, and the cytotoxic activity of the NS2 cell line against its LCL stimulator cell line was diminished in the presence of monoclonal antibodies specific for CD56 (*Suzuki et al., 1991*). Conversely, the expression of NCAM on some target cell lines inhibits NK cell-mediated cytotoxicity (*Jarahian et al., 2007*). Here, using CD56-deficient NK92 and YTS human NK cell lines, we demonstrate a requirement for CD56 in killing of CD56-negative targets by human NK cell lines and define a mechanism for the requirement for CD56 in human NK cell cytotoxicity.

Our previous study defined a role for CD56 in human NK cell migration, and we had previously modeled this requirement using the NK92 cell line (*Mace et al., 2016*). Here we extended our studies to the use of YTS cells as a second human NK cell line. Despite a significant and consistent effect of CD56 deletion on the cytotoxic function of NK92 cells, this effect was not conserved between the two human NK cell lines that we tested. The profound deficiency in lytic function in the NK92 cell line was observed even when effector cells were incubated with targets for just one hour in a $^{51}$Cr release assay, a timescale which likely only permits a single target cell killing to occur (*Gwalani and Orange, 2018*). As this incubation time led to significant killing of target cells by WT NK92 cells, these data suggest that the observed decrease in killing by the NK92 CD56-KO cells is not due to an inability to mediate serial killing against multiple targets despite the previously demonstrated requirement for CD56 in NK92 cell migration (*Mace et al., 2016*; *Bhat and Watzl, 2007*). Furthermore, our subsequent recapitulation of cytotoxicity using antibody-coated glass to activate NK cells uncoupled the defect that we describe in exocytosis from one of cell migration.

For YTS cells, killing was only impaired at the 1 hr time point, signifying an initial delay in their ability to kill which could be rescued at later time points. The milder effect on cytotoxicity in the YTS cell line when compared to the NK92 cell line suggests fundamental differences in the requirements of the two cell lines for cytolytic function, and it remains possible that the effect on YTS cell cytotoxicity could be due to impaired cell migration affecting serial killing of targets at later time points given the requirement for CD56 in human NK cell migration (*Mace et al., 2016*). However, impaired IFNγ production by YTS CD56-KO cells following contact-dependent co-incubation with 721.221 target cells further underscores underlying differences in the requirement for CD56 in these cell lines and also demonstrates a functional relevance for CD56 in the YTS cell line. Previously demonstrated phenotypic and genotypic differences between the cell lines (*Gunesch et al., 2019*) supports this hypothesis and suggests that further investigation is required to fully delineate the relative role of CD56 in these contexts. Here, we have chosen to focus on the mechanism underlying the requirement for CD56 in the cytotoxic function of NK92 cells.

Our investigations into the mechanism of CD56 function in human NK cells were informed by extensive investigations into the functional role of NCAM in neural cells, where NCAM-mediated signaling plays a critical role in axonal growth, survival and proliferation (*Ditlevsen et al., 2008*). While there are multiple pathways by which this can occur that include homotypic and heterotypic NCAM interactions, NCAM signaling includes the formation of complexes that signal through Ras-MAPK-ERK pathways via interactions with p59$^{fyn}$ and FAK. Specifically, NCAM140 constitutively interacts with p59$^{fyn}$, whereas FAK is recruited upon cross-linking of the NCAM extracellular domain (*Beggs et al., 1997*). Recruitment of FAK leads to its phosphorylation and kinase function, and ultimately growth cone migration. The demonstrated requirement for Src family kinases, including Fyn, in NK cell cytotoxicity suggested that a common pathway could be functioning in NK cells (*Dong et al., 2012*). Furthermore, while FAK is not known to be expressed in human NK cells, Pyk2 is expressed and is required for polarization of the MTOC during IS formation and resultant cytotoxic function (*Sancho et al., 2000*). Here, we define a requirement for CD56 in phosphorylation of Pyk2 on Y402, the primary autophosphorylation site that serves as a docking site for the SH2 domain of Src family kinases (*Eide et al., 1995*). This requirement was most significant in the NK92 cell line, which we found had a significantly greater amount of Pyk2 Y402 phosphorylation both at baseline and upon activation. The functional implication of this finding was the significantly greater impairment in NK92 cytotoxicity following pre-treatment of both cell lines with a Pyk2 inhibitor.

Despite showing that Pyk2 is required for NCAM-mediated function, it is still unclear precisely the mechanism by which this signaling occurs. We demonstrate that loss of CD56 leads to abolishment of lytic granule exocytosis, as surface CD107a was decreased on NK92 CD56-KO cells compared WT NK92 after activation with either plate-bound antibodies or with target cells. In addition, the

maximum activity of granzyme A in the supernatant, an indirect readout of exocytosis, was decreased for NK92 CD56-KO cells. Given that the exocytosis defects we observed in the NK92 cell line were demonstrated using antibody cross-linking, and thus independently of target cell activation, it is unlikely that CD56 is binding directly to unknown target cell ligands in this context. Furthermore, our use of CD56-negative target cells defines independence from homotypic NCAM interactions. NCAM can also bind both in cis and in trans to FGFR, and interactions between CD56 on NK cells and FGFR1 on T cells is sufficient to induce T cell IL-2 production (*Kos and Chin, 2002*). Despite this indication that productive interactions occur within immune cells upon CD56-FGFR1 interactions, we have repeatedly failed to detect FGFR1 on human NK cells or the target cells used in this study (data not shown), suggesting that FGFR1 is not the mechanism by which NK cell function is mediated by CD56.

Given the defect in exocytosis in CD56-deficient NK92 cells, we sought to define earlier steps in NK cell cytotoxicity and found that actin accumulation at the synapse was decreased in CD56-KO cells, although not significantly, and that MTOC polarization was impaired. While these features could be attributed to impaired integrin-mediated adhesion, conjugation of effectors to targets was not impaired in CD56-KO cells. Given the previously described requirement for Pyk2 in MTOC polarization during NK cell cytotoxicity (*Zhang et al., 2014*; *Sancho et al., 2000*), these data further suggest that the requirement for CD56 is in Pyk2-mediated function during cytotoxicity. This was further defined by impaired Pyk2 phosphorylation and recruitment to the IS in NK92 CD56-KO cells. Taken together, we propose a mechanism by which CD56 helps to localize or retain Pyk2 at sites of potential activation, where it could be available for subsequent autophosphorylation and activation of downstream signaling pathways required for polarization. Such a model has been proposed as the mechanism by which NCAM signals through FAK (*Beggs et al., 1997*). In addition, the well-defined requirement for activating microcluster formation in activation at the immune synapse is mediated by adhesion receptors, and integrin engagement promotes microcluster formation, mobility and signaling at the NK cell IS (*Steblyanko et al., 2015*). Inhibition of Pyk2 phosphorylation abrogates this effect, underscoring the importance of Pyk2 in amplifying activating signaling that is promoted by integrin engagement (*Steblyanko et al., 2015*). Here, we show another example of how an adhesion molecule may be playing a similar role; whether this is through direct binding of Pyk2 to CD56 through Fyn, or through indirect mechanisms, such as regulation of integrin spatial localization, remains to be defined. There are other mechanisms that could be at play, including the previously described role for NCAM in regulating lipid raft inclusion and exclusion of signaling receptors (*Niethammer et al., 2002*). Finally, given the significant degree of polysialation of CD56 on NK cells, particularly NK92 cells, the steric hindrance and negative charge of polysialated CD56 may be modulating membrane dynamics by physically influencing the organization and activity of integrins and activating receptors (*Paszek et al., 2014*). While the mechanisms of the regulation of this signaling remain to be elucidated, our data speak to CD56 being an important player in the orchestration of adhesion and cellular signaling at the NK cell membrane. Delineating the role that CD56 is playing in directly mediating signaling, as opposed to spatially modulating other players such as integrins and activating receptors, represents an exciting direction for NK cell biology.

While demonstrating a requirement for CD56 in cytotoxicity mediated by NK92 cells, we found that primary NK cells mediated target cell killing when CD56 was deleted from either total NK cells or specifically from the CD56[bright] subset. Given the challenges inherent in manipulating gene expression in primary cells, we were reliant upon the use of cytokines, in this case IL-15, to promote NK cell survival and expansion following gene editing. It remains unclear as to whether this expansion bypasses a requirement for CD56 function in primary NK cells, although we do demonstrate that IL-15 stimulation of ex vivo NK cells significantly reduces reliance on Pyk2 Y402 phosphorylation for NK cell lytic function based upon the effect of a Pyk2 inhibitor. Despite the apparent reduced dependence on Pyk2 phosphorylation in IL-15 activated primary cells and the robust phosphorylation of Pyk2 Y402 in NK92 cells, activation of the NK92 cell line with IL-15 failed to rescue the defect in cytotoxicity in NK92 CD56-KO cells. This, combined with the observation that Pyk2 inhibition does not abrogate NK92 lytic function to the same extent as CD56 deficiency, suggests that there may be other mechanisms by which CD56 is exerting function in human NK cells. Ultimately, we show that despite the requirement for CD56 function in cell lines, ex vivo human NK cells are independent of this requirement, a finding that has relevance for clinical applications of NK cell-mediated adoptive therapies.

Consistent with previous reports (*Lanier et al., 1991*), our data demonstrate that the transmembrane-containing 140 kDa NCAM (CD56) isoform is primarily expressed on human NK cells. Western blot analyses demonstrated a molecular weight range of ~130–160 kDa, which was significantly decreased following treatment of NK cell lines with PNGase F to remove polysialic acid. While these experiments, which led to the generation of a band that appeared close to 120 kDa, suggested that NCAM-120 could be expressed intracellularly (*Drake et al., 2008*), treatment with PI-PLC, which we demonstrated cleaved GPI-anchored proteins, did not affect the detection of CD56 on the cell surface. The single band that we detected was also consistent with the predicted weight of the NCAM140 isoform containing 858 amino acids (*Lanier et al., 1989*). Furthermore, expression of the extracellular domain (ΔICD) or the intracellular domain (ΔECD) of CD56 failed to restore cytotoxic function to the NK92 CD56-KO, whereas expression of full-length 140 kDa NCAM was sufficient. These experiments also concur with previous evaluations of transcript level expression that also describe the predominant isoform in human PBMCs as the 140 kDa isoform (*Lanier et al., 1989*). While other reports have suggested that NCAM120 is the primarily expressed isoform on freshly isolated NK cells based upon qPCR (*Van Acker et al., 2019*), the use of Western blotting in our study allows us to directly assess protein expression. The degree of polysialation in human NK cells is notable, however, particularly in NK92 cells and primary NK cells, and this high level of polysialation in human NK cells suggests that polysialic acid itself may be playing a role in CD56 function through its steric properties.

In summary, here we describe a novel, functional role for CD56, the canonical identifier of human NK cells. We build upon previous studies describing a role for CD56 in NK cell cytotoxicity and those that define its role in cell migration and NK cell maturation to elucidate, for the first time, its role in the lysis of CD56-negative targets. In doing so we describe an unexpected role for CD56 in the function of commonly used NK cell lines, including NK92 cells, but demonstrate that primary NK cells can execute cytotoxicity independently of CD56. Finally, we link the cytotoxicity defect in CD56-KO NK92 cells to impaired signaling through the non-receptor tyrosine kinase Pyk2, and thus define novel roles for CD56 and Pyk2 in human NK cell cytotoxic function.

## Materials and methods

Additional details on key reagents and resources are included in the key resources table (*Supplementary file 1*).

### Human NK cells

For imaging experiments and cytotoxicity assays where indicated, primary human NK cells were isolated from healthy donors by venipuncture followed by NK cell enrichment using RosetteSep (Stemcell Technologies) and separation by Ficoll-Paque density gradient. Cells were resuspended in complete R10 media then co-cultured with K562 target cells and prepared for fixed cell confocal imaging as described below. Primary cells for CRISPR-Cas9 deletion experiments were obtained from leukopaks obtained from anonymous healthy platelet donors and enriched using RosetteSep and Ficoll-Paque Plus according to manufacturer's instructions. Where indicated, primary NK cells were cultured in complete R10 media containing 3 ng/ml IL-15. Primary NK cells were obtained in accordance with the Declaration of Helsinki with the written and informed consent of all participants under the guidance of the Institutional Review Boards of Baylor College of Medicine and Columbia University.

### Cell culture and generation of CD56-KO cell lines

NK cell lines and target cell lines were maintained at an approximate concentration of $1–5 \times 10^5$ cells/ml and cultured at 37°C 5% $CO_2$. YTS, Jurkat, Raji, 721.221, K562 and KT86 cells (*Banerjee et al., 2007*) were a gift from Dr. J. Orange (Columbia University) and NK92, Phoenix and Jurkat cells were acquired from ATCC. RPMI supplemented with 10% fetal bovine serum (FBS) plus essential nutrients was used to culture YTS, Jurkat, Raji, 721.221, K562 and KT86 cells. NK92 cell lines were maintained in Myelocult media (Stemcell Technologies) supplemented with 200 U/ml of IL-2 (Roche) and 50 U/ml of penicillin/streptomycin (Gibco). All cell lines were confirmed to be mycoplasma negative every 3 months and cell line identify was routinely confirmed by flow cytometry assessing expression of known surface markers. WT NK92 and YTS cells were validated by their

expression of CD56 and absence of CD3 expression, combined with expression of CD28 (YTS only) or CD2 (NK92 only). CD56-KO cells were similarly validated but excluding consideration of CD56 expression. Jurkat cells were confirmed to be CD55[+] and Raji cells were confirmed to be CD20[+] by flow cytometry.

*NCAM1* (CD56) was deleted in YTS using CRISPR gene editing as previously described for our generation of the NK92 cell line (*Mace et al., 2016*). In brief, a U6gRNA-Cas9-2A-GFP vector (Sigma-Aldrich) containing a *NCAM1*-specific gRNA sequence was incorporated into the NK cell lines through nucleofection. $10^6$ cells per condition were nucleofected with 4 µg of DNA using the Amaxa Nucleofector II (Lonza, Kit R; program R-024). Nucleofected cells were allowed to recover for 24–48 hr before sorting for GFP positivity and lack of CD56 expression.

NCAM reporter plasmids were generated by Epoch Life Sciences Inc and were made by subcloning NCAM1 (NCBI reference sequence: NM_000615.6, transcript variant 1; Origene) into BamHI and SalI digested pBABE-puro-mApple retroviral plasmid. For construction of chimeric constructs, inserts were assembled by PCR and cloned into pBABE-puro-mApple between BamHI and SalI using ligation independent cloning by Epoch Life Sciences Inc. Re-expression of CD56 or expression of chimeric constructs in NK92 CD56-KO cell lines was performed by retroviral transduction. Phoenix-AMPHO cells (ATCC) were transfected with 4.5 µg of plasmid DNA using Fugene 6 Transfection Reagent (Promega). Supernatant containing viral particles was collected and concentrated using PEG-IT (System Biosciences) followed by centrifugation at 2600 rpm. NK92 CD56-KO cells were transduced with viral particles in the presence of MAX Enhancer and TransDux (System Biosciences). Within 48 hours cells were sorted for CD56 and mApple and cells were expanded and maintained under antibiotic selection with routine phenotypic validation.

## Primary NK cell electroporation

Primary human NK cells were incubated after enrichment for 16 hr at 37°C in 3 ng/mL IL-15 (Miltenyi) in RPMI media supplemented with 10% Human AB serum (Sigma). Cells were washed with PBS, twice, and suspended at $2 \times 10^7$ cells/ mL in EP Buffer (MaxCyte). Using OC-100 processing assemblies and the MaxCyte GT (Maxcyte), cells were electroporated in the presence of Cas9 mRNA (Tri-Link) and CD56 guide RNA (CGCUGAUCUCCCCCUGGCU; IDT) using the WUSTL-3 setting and transferred to a 12-well plate and allowed to rest for 10 min at 37°C (*Cooper et al., 2018*). Pre-warmed media containing 3 ng/mL IL-15 was added to the cells which were then cultured in IL-15 containing media as indicated. Cells were sorted based on CD56 expression on BD FACS Aria II to >90% purity. Flow-based killing assays were performed for 1 hr using CFSE-labeled K562 as previously described (*Leong et al., 2014*).

## Chromium release assay

NK cell effector cells were co-cultured with target cells that had been pre-incubated with 100 µCi $^{51}$Cr for 1 or 4 hr in 96-well round-bottomed plates at 37°C 5% $CO_2$. 1% IGEPAL (v/v) (Sigma-Aldrich) was used to lyse maximal release control wells and plates were centrifuged. Supernatant was transferred to a LUMA plate (Perkin Elmer) and dried overnight. Plates were read with a TopCount NXT and % specific lysis was calculated as follows: (sample – average spontaneous release) / (average total release – average spontaneous release) x 100. For experiments done with Pyk2 inhibition, effectors were pre-incubated for 10 min with 5 µM PF431396 (Tocris) then assays were performed in the presence of 5 µM PF431396 or equivalent volume of DMSO as a vehicle control.

## BLT esterase assay

96-well flat-bottomed plates were pre-coated overnight at 4°C with 5 µg/ml anti-NKp30 (clone P30-15, Biolegend) and -CD18 (clone IB4) or with mouse IgG1aκ as an isotype control. Wells were blocked with phenol red-free RPMI complete medium then washed three times with PBS. $10^5$ WT or CD56-KO NK92 cells were added per well. Plates containing cells were briefly centrifuged then incubated for 90 min at 37°C 5% $CO_2$. Following incubation 1% v/v IGEPAL Sigma-Aldrich) was added to maximum release wells and plates were centrifuged at 1000 rpm. Supernatant was transferred and substrate solution containing PBS, HEPES (Gibco), N-α-Cbz-L-lysine thiobenzyl ester hydrochloride (BLT; Sigma-Aldrich) and 5,5'-Dithiobis(2-nitrobenzoic acid) (DTMB; Sigma-Aldrich) was added. The plate was incubated at 37°C 5% $CO_2$ for 30–60 min and luminescence was read at 415 nm using the

BioTek Synergy H4 Hybrid plate reader. % maximum activity was calculated as: (sample absorbance – average background)/(average total release – average background) x 100%.

## CD107a degranulation assay and detection of phosphorylated proteins by flow cytometry

NK92 and YTS cell lines were assessed for surface CD107a expression as a marker for degranulation after activation. $2 \times 10^5$ WT or CD56-KO NK92 and YTS cell lines were added to 12-well flat-bottomed plates that were pre-coated overnight at 4°C with 5 µg/ml anti-NKp30 (clone P30-15, Biolegend) and -CD18 (clone IB4) or 5% bovine serum albumin as a negative control. Alternatively, YTS cells were cocultured with $10^5$ 721.221 target cells at a 2:1 effector to target ratio in 5 ml polystyrene round-bottom tubes. Each condition was run in triplicate. Anti-CD107a antibody (eBioscience, clone H4A3) was added at the onset of incubation for co-culture experiments. Cells added to the pre-coated antibody plates were spun briefly at 200 rpm before incubation for 90 min at 37°C 5% $CO_2$. Cells activated by plate-bound antibodies were mechanically dislodged and transferred to 5 ml polystyrene round-bottom tubes and incubated for an additional 25 min with anti-CD107a. Cells were fixed using 300–400 µl of 2% paraformaldehyde (Electron Microscopy Sciences), then CD107a surface expression was measured using a modified LSR Fortessa (BD Biosciences) with the CD107a$^+$ gate defined by unstained negative control cells. Data were analyzed using FlowJo 10 (BD Biosciences). For experiments performed with target cell activation the average background of media alone was subtracted from samples.

For detection of phospho-Pyk2, cells were activated on plates as above and then fixed and permeabilized with BD Cytofix/Cytoperm (BD Biosciences) followed by detection with pPyk2 Y402 (Abcam, ab4800, 1:50) and secondary detection with goat anti-rabbit IgG Alexa Fluor 488 (Invitrogen, 1:100). Mean fluorescence intensity was adjusted to relative fluorescent intensity by normalizing to the WT condition.

## Flow cytometry

FACS analysis of WT or CD56-KO NK92 and YTS cell lines was performed using modified NK cell panels designed to assess the expression of adhesion, inhibitory, and developmental ligands and intracellular molecules (*Mahapatra et al., 2017*). For panels evaluating response to activation 10 ng/ml phorbol 12-myristate 13-acetate (PMA, Sigma-Aldrich) and 1 µg/ml of ionomycin were used to stimulate cells for 4 hr at 37°C 5% $CO_2$. 10 µg/ml of Brefeldin A (Sigma-Aldrich) was added at the onset of incubation to stimulated and unstimulated controls to prevent protein transport of newly synthesized proteins in response to cellular activation. After incubation, activated cells were stained for surface ligands for 25 min followed by permeabilization and fixation using BD Cytofix/Cytoperm (BD Biosciences). Antibody staining was performed in the dark and at room temperature; cells were then stained for intracellular markers for 1 hr before being washed and resuspended in PBS 1% paraformaldehyde (Sigma-Aldrich). Cells in panels without PMA/ionomycin stimulation were also resuspended in PBS 1% paraformaldehyde after being incubated with antibodies for 25 min.

Detection of CD56 on cell lines was performed using clone HCD56 (1:100, Biolegend) and polysialated NCAM (PSA-NCAM) was detected by clone 12F8 (BD Biosciences) followed by anti-rat goat IgG FITC (Invitrogen). $10^4$ events per sample were acquired using a modified LSR Fortessa (BD Biosciences). Fluorescence minus one controls (FMO) were used to define positive and negative populations. Prism 6.0 (GraphPad Software) was used to graph the percent positive and the mean fluorescence intensity calculated using FlowJo 10 (BD Biosciences) software.

For measurement of cell conjugates, NK cells and targets were pre-incubated with eFluor 670 (Thermo) and Cell Tracker Green (Thermo) respectively, then $10^5$ cells were co-incubated at a 1:1 effector:target ratio in 5 ml polystyrene FACS tubes for 0, 10, 30, 60 or 120 min. At the indicated timepoints 2% paraformaldehyde was added to the tubes to fix conjugates then all samples were analyzed on a BD Fortessa flow cytometer. The frequency of cells in conjugates was calculated using FlowJo 10 (BD Biosciences) and data were plotted and tested for statistical significance using Prism 8.0 (GraphPad Software). The mean of each triplicate condition was calculated, and these means were compared by rank sum (Mann-Whitney) test for four independent technical replicates.

## IFNγ detection by ELISA

WT YTS and CD56-KO cells were incubated with 721.221 target cells at a 2:1 ratio at 37°C 5% $CO_2$ in round-bottomed 96-well plates for 22 hr in triplicate. Plates were centrifuged and the supernatant collected. Human IFNγ was detected by ELISA (Abcam) following the manufacturer's procedure. Absorbance was read at 415 nm on a BioTek Synergy H4 Hybrid plate reader. IFNγ concentrations were calculated using a standard curve following subtraction of background (media only) from all conditions. Results were graphed and statistical analyses were performed using Prism 6.0 (GraphPad software).

## PI-PLC cleavage of GPI-anchored proteins and detection by flow cytometry

NK92 and YTS NK cells, or Raji and Jurkat cells as a positive control were incubated with 1 U/ml of phosphatidylinositol-specific phospholipase C (PI-PLC; Invitrogen) at 4°C for 30 min in cold PBS. Cells were then washed twice with cold PBS and transferred to polystyrene tubes. Cells were immunostained for 25 min at room temperature with anti-CD56 (BV421; clone HCD56, Biolegend) or anti-CD55 (PE; clone JS11, Biolegend). Cells were washed once with PBS and then resuspended in 2% paraformaldehyde and analyzed by flow cytometry. Prism (GraphPad Software) was used to graph the percent positive and the mean fluorescence intensity calculated using FlowJo 10 (BD Biosciences) software.

## Lysate preparation and western blots

Cell lysates from 5 to $10 \times 10^6$ cells were generated using CHAPS Cell Extract Buffer (Cell Signaling Technology) supplemented with 1X Halt protease inhibitor cocktail (Thermo Fisher Scientific). Samples were incubated at 95°C with NuPAGE sample reducing agent (Thermo Fisher Scientific) and 4X NuPAGE LDS sample buffer (diluted to 1X) for 10 min. $2-4 \times 10^5$ cell equivalents per well were loaded into a NuPAGE 4–12% Bis-Tris density gradient gel (Thermo Fisher Scientific) and ran at a constant 150V for 80 min. Separated proteins were transferred onto nitrocellulose membranes using a Mini Gel Tank/Mini Blot Module (Life Technologies) at a constant 0.2A for 105 min. The nitrocellulose membranes were then blocked with 5% nonfat milk in PBS 0.05% Tween-20 for 90–120 min at 4°C. Nitrocellulose membranes were incubated overnight at 4°C with primary antibodies in 5% (w/v) BSA in PBS 0.05% Tween 20 at the following dilutions: 1:1000 anti-CD56 (mouse monoclonal, clone 123C3; Cell Signaling Technology) and 1:4000 anti-actin (rabbit polyclonal; Sigma-Aldrich) as a loading control. Detection of polysialated NCAM (PSA-NCAM) was by clone 2-2B (Millipore, 1:1000). Membranes were washed with 0.5M NaCl in PBS 0.05% Tween 20. Primary antibodies were probed with either IRDye 680RD goat anti-mouse IgG or IRDye 800CW goat anti-rabbit IgG (Li-COR Biosciences, 1:10,000) secondary antibodies for 1 hr at room temperature. Nitrocellulose membranes were imaged using the Odyssey CLx imaging system (Li-COR Biosciences). Image Studio Lite software was used for densitometry and analysis of Western blot images.

## Confocal microscopy

For fixed cell imaging, WT and CD56-KO NK92 cells were co-cultured with K562 target cells at a 2:1 effector to target ratio in complete R10 medium. Cells were incubated for 20 min at 37°C 5% $CO_2$ then were transferred to poly-L-lysine coated #1.5 coverslips for an additional 25 min. Following incubation, cells were fixed and permeabilized with CytoFix/CytoPerm (BD Biosciences) at room temperature for 15 min and were washed twice with 50–100 μl of PBS. Conjugate immunostaining was performed with biotinylated monoclonal mouse anti-tubulin (Invitrogen) and Brilliant Violet 421-conjugated streptavidin (Invitrogen); Alexa Fluor 488-conjugated mouse anti-perforin (clone dG9); phalloidin Alexa Fluor 568; pPyk2 Y402 (Abcam, ab4800) and goat anti-rabbit IgG Alexa Fluor 488 (Invitrogen). Coverslips were mounted to slides using ProLong Gold antifade reagent (ThermoFisher Scientific). For detection of CD56 in conjugates, NK cells were pre-incubated with HCD56 Alexa Fluor 647 (Biolegend) prior to conjugate formation. For experiments done with Pyk2 inhibition, effectors were pre-incubated for 10 min with 5 μM PF431396 (Tocris) then assays were performed in the presence of 5 μM PF431396 or equivalent volume of DMSO as a vehicle control. Images were acquired through a 63 × 1.40 NA objective on a Zeiss AxioObserver Z1 microscope stand equipped with a Yokogawa W1 spinning disk. Illumination was by solid state laser and detection by Prime 95B

sCMOS camera. Data were acquired in 3i Slidebook software and exported as TIFF files for further analysis.

## Image analyses

Fiji (*Schindelin et al., 2012*), Volocity (Perkin Elmer) or Imaris (Bitplane) were used to process and analyze confocal image sequences. MTOC polarization to the synapse was determined using the line measurement tool after denoting the highest point of fluorescence intensity of α-tubulin as the MTOC. For actin accumulation at the synapse, the stamp tool was used to measure the fluorescence intensity of actin at the synapse and distal region of both the effector and target cells (*Banerjee and Orange, 2010*). The background and initial synapse intensity (AU) of actin was calculated as the area ($\mu m^2$) x the mean of fluorescence intensity. Total intensity of actin at the synapse was calculated as follows: Total synapse intensity = synapse intensity – (effector background intensity + target background intensity). Lytic granule convergence was calculated by measuring the distance from individual granules to the MTOC as defined as the brightest point of α-tubulin intensity (*Hsu et al., 2017*).

For the accumulation of pPyk2 Y402 and CD56 at the synapse of primary NK cells, masks of synaptic vs. non-synaptic actin were generated following auto thresholding of intensity in the actin channel. Intensity of pPyk2 Y402 or CD56 staining was measured in Fiji following auto thresholding, with the 'limit to threshold' box checked. Intensity of respective channels of interest at the synapse or the non-synaptic cortical region were plotted for individual cells. All microscopy experiments were repeated by multiple individuals and in some cases masking of experimental conditions was performed.

## Statistical analyses

Prism 6.0 or 8.0 (GraphPad software) was used for statistical analyses. Unless otherwise stated, analyses used either a Student t-test with Welch's correction or a one-way ANOVA with multiple comparisons. Data with non-normal distribution were tested with a Mann-Whitney (rank sum) test. Statistical significance was denoted when differences in measurements produced a p-value of $<0.05$. Sample size computations were not performed but experiments with cell lines were replicated at least three times on different days using different passages of cell lines to generate technical replicates of at least three for each experiment. Experiments with ex vivo NK cells were performed with at least three biological and technical replicates.

## Acknowledgements

This work was supported in part by R01AI137073 to EMM. The authors have no competing interests to declare. We thank the Genome Engineering and iPSC Center (GEiC) at the Washington University in St. Louis for gRNA validation services.

## Additional information

### Funding

| Funder | Grant reference number | Author |
| --- | --- | --- |
| National Institutes of Health | R01AI137073 | Emily M Mace |
| National Institutes of Health | R01CA205239 | Todd A Fehniger |
| National Institutes of Health | P50CA171063 | Todd A Fehniger<br>Melissa M Berrien-Elliott |
| National Institutes of Health | K12CA167540 | Melissa M Berrien-Elliott |

The funders had no role in study design, data collection and interpretation, or the decision to submit the work for publication.

### Author contributions

Justin T Gunesch, Data curation, Formal analysis, Investigation, Writing - original draft, Writing - review and editing; Amera L Dixon, Tasneem AM Ebrahim, Melissa M Berrien-Elliott, Swetha

Tatineni, Everardo Hegewisch-Solloa, Formal analysis, Investigation, Writing - original draft, Writing - review and editing; Tejas Kumar, Formal analysis, Investigation; Todd A Fehniger, Conceptualization, Supervision, Writing - original draft, Writing - review and editing; Emily M Mace, Conceptualization, Formal analysis, Supervision, Funding acquisition, Investigation, Writing - original draft, Writing - review and editing

### Author ORCIDs
Todd A Fehniger ID http://orcid.org/0000-0002-8705-2887
Emily M Mace ID https://orcid.org/0000-0003-0226-7393

### Ethics
Human subjects: Peripheral blood NK cells were obtained in accordance with the Declaration of Helsinki with the written and informed consent of all participants under the guidance of the Institutional Review Boards of Baylor College of Medicine (IRB H-30487) and Columbia University (IRB AAAR7377).

### Decision letter and Author response
Decision letter https://doi.org/10.7554/eLife.57346.sa1
Author response https://doi.org/10.7554/eLife.57346.sa2

## Additional files

### Supplementary files
- Supplementary file 1. Key resources table.

- Transparent reporting form

### Data availability
All data generated or analysed during this study are included in the manuscript and supporting files.

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
