## [Decision Letter]

**Acceptance summary:**

The demonstration of the role of CD56 and Pyk2 in NK cell killer and polarisation is clear and important. Your transparency in presenting additional data on the non-essential nature of the pathway in other cellular contexts is also appreciated, but doesn't depreciate the value of the positive results in opening this to further investigation. The work will be of interest to immunologists based on adding to established and new roles of CD56 and Pyk2 and also neurobiologists based on desire to understand functional potential of CD56/NCAM in different synaptic contexts.

**Decision letter after peer review:**

Thank you for submitting your article "CD56 regulates human NK cell cytotoxicity through Pyk2" for consideration by *eLife*. Your article has been reviewed by three peer reviewers, including Michael L Dustin as the Reviewing Editor and Reviewer #1, and the evaluation has been overseen by Anna Akhmanova as the Senior Editor.

The reviewers have discussed the reviews with one another and the Reviewing Editor has drafted this decision to help you prepare a revised submission.

Your study demonstrates that 1) CD56 is important in the NK92 cell lines and to some extent the YTS line, 2) this pathway is not required in primary NK cells, 3) reduced capacity to activate Pyk2 appears to play a role in the dysfunction in NK92 cells. The findings demonstrate an important pathway, but the results can be better presented and some additional information is needed to interpret the results.

Essential revisions:

1) The Introduction could be expanded to better discuss the role of other adhesion molecules, particularly integrins that could also be involved in activation of Fyn and Pyk2. Also, you should look at the work on M. Sixt on the role of PSA in activation of CCL21. This might be cited as another example of a PSA core protein (CCR7) and might provide another functional angle on role of CD56 in NK cell migration/interactions based on activation of CCL21 by PSA on CD56 that might be cross-presented to cells expressing CCR7. Although this is outside the scope of this paper, it should be kept in mind as there are few situations in biology where PSA is invoked.

2) Different expression levels of NKp30 and CD18 on NK cell lines could significantly influence the experiments involving stimulation with anti-NKp30/CD18. You should provide surface staining on YTS and NK92 in WT, KO and rescued variants (when available) should be shown in a way that allows quantitative comparisons.

3) CD56-KO disrupts migration in your previous work and Pyk2 is known from other work to be key for integrin signalling. But you have not formally addressed whether reduced adhesion is playing a key role in the defects observed. You should perform a flow cytometry-based adhesion time-course of differentially fluorescently labelled NK and target cells. It is possible that this could help explain the difference among the cell lines and primary cells.

4) Although evidence is provided that cytolytic activity in NK92 cells is granule-mediated, however, classical experiment with EGTA added to extracellular medium that chelates Ca^2+^ precluding calcium influx, which is necessary for granule delivery and release would provide unambiguous evidence of granule-mediated killing. This experiment would be particularly important to do for YTS cells that killed the same target cell very well but did not release detectable amount of cytolytic granules.

---

## [Author Response]

Essential revisions:1) The Introduction could be expanded to better discuss the role of other adhesion molecules, particularly integrins that could also be involved in activation of Fyn and Pyk2. Also, you should look at the work on M. Sixt on the role of PSA in activation of CCL21. This might be cited as another example of a PSA core protein (CCR7) and might provide another functional angle on role of CD56 in NK cell migration/interactions based on activation of CCL21 by PSA on CD56 that might be cross-presented to cells expressing CCR7. Although this is outside the scope of this paper, it should be kept in mind as there are few situations in biology where PSA is invoked.

We thank the reviewer for raising this relevant point and have included a broader discussion of the relationship between integrins, Pyk2 and Fyn (and their downstream signaling) in the Introduction. We have also included a brief description of the study describing PSA on CCR7 in the Introduction, as this reference makes an important point about the relevance of PSA in innate immune function. As the reviewer points out, the unusual nature of PSA modifications makes this a very interesting mechanism, and while beyond the scope of the current study we are very interested in the relatively high levels of PSA found on human NK cells and the significance of this modification on cell migration and signaling.

2) Different expression levels of NKp30 and CD18 on NK cell lines could significantly influence the experiments involving stimulation with anti-NKp30/CD18. You should provide surface staining on YTS and NK92 in WT, KO and rescued variants (when available) should be shown in a way that allows quantitative comparisons.

We thank the reviewer for suggesting this important control experiment. While we had addressed the expression of CD18 in WT and CD56-KO NK92 and YTS cell lines in our original submission (Figure 2—figure supplement 2), NKp30 was not included in our panel and the rescued variants were not included in our analyses. We propose performing surface staining of all cell lines for NKp30 and CD18 upon our return to the lab and updating the existing Figure 2—figure supplement 2 to include these data, as well as modifying the text at that time to reflect the results of these experiments.

3) CD56-KO disrupts migration in your previous work and Pyk2 is known from other work to be key for integrin signalling. But you have not formally addressed whether reduced adhesion is playing a key role in the defects observed. You should perform a flow cytometry-based adhesion time-course of differentially fluorescently labelled NK and target cells. It is possible that this could help explain the difference among the cell lines and primary cells.

We thank the reviewer for the opportunity to address this important question. Given the known role for Pyk2 in integrin mediated adhesion and signaling, we have considered that adhesion to target cells mediated by LFA-1 may be negatively affected in the NK92 CD56-KO cells. As we reflected on this during the course of the study, we actually have existing data, generated prior to our laboratory shutdown, that addresses this question and indicates that adhesion of CD56-KO NK92 cells to targets is not impaired. We have included these existing data as a new Figure 4E. Using flow cytometry-based conjugation assays, as the reviewer suggests, we found that there was a surprising, yet significant and consistent, increase in the frequency of NK92 CD56-KO cells conjugated to K562 target cells relative to WT cells at 30 and 60 minutes, whereas the difference between WT and CD56-KO cells is less pronounced at the 120 minute timepoint and not detected at the 10 minute timepoint. Upon performing these experiments, we were satisfied that at least initial adhesion to targets is not impaired, although there may be additional aspects of outside-in and inside-out integrin regulation that remain to be uncovered that are leading to apparent increased target formation by CD56-KO cells at the middle time points of the assay. One interpretation of these data could also be that disengagement from target cells is impaired, which led us to think that serial killing could be affected and ultimately led us to perform ^51^Cr assays at one hour in addition to the 4-hour. However, the similarly impaired cytotoxicity at the 1-hour timepoint in CD56-KO cells shown in Figure 2C has prompted us to conclude that it is not impaired disengagement from targets that is responsible for the cytotoxic defect that we see at 4 hours.

While we conclude from these data that CD56-KO cells are not impaired in their ability to adhere to targets, the altered kinetics of target cell binding suggested by our adhesion assays and the migration defects that remain incompletely defined suggest that we still have much to learn about the molecular interactions between CD56, Pyk2 and integrins. While these studies are beyond the scope of the current work, molecular dissections of these signaling pathways are ongoing investigations in the laboratory.

4) Although evidence is provided that cytolytic activity in NK92 cells is granule-mediated, however, classical experiment with EGTA added to extracellular medium that chelates Ca^2+^ precluding calcium influx, which is necessary for granule delivery and release would provide unambiguous evidence of granule-mediated killing. This experiment would be particularly important to do for YTS cells that killed the same target cell very well, but did not release detectable amount of cytolytic granules.

We agree with the reviewer that the question of whether YTS cells are using a granule-independent pathway of target cell killing is a critical point. We had considered this point during the course of our study and had approached it in two indirect ways, although the data was not shown in our original submission. The first approach was to measure expression of FASL and TRAIL on YTS cells, and we did not detect significant up-regulation of these receptors that can mediate granule-independent target cell lysis. We additionally evaluated data from perforin-deficient WT YTS cell lines and found that these cells failed to kill target cells. Together, these data led us to preliminarily conclude that it was not likely that the YTS cells were compensating with perforin-independent pathways of killing.

Despite both these experiments suggesting that YTS killing is dependent upon lytic granule delivery, they do not directly address the question specifically in the context of the CD56-KO cell lines. We agree that the experiment proposed by the reviewer, in which we chelate Ca++ to prevent calcium-dependent exocytosis, is a much better way to address the finding that NK92, but not YTS cells, are dependent on CD56 for target cell lysis. Importantly, the proposed approach will allow us to interrogate both WT and CD56-KO NK92 and YTS cells using a similar platform. We propose that we perform these experiments when we have access to our laboratory, using ^51^Cr cytotoxicity assays as a measurement of NK cell lytic function, and incorporate the results of these experiments into our bioRxiv preprint or as a Research Advance. We appreciate the suggestion from the reviewer and feel that the addition of these data, regardless of the outcome of our findings, will significantly strengthen our manuscript. To be fully responsive to the reviewer comment at this time we have included a discussion of the importance of such an experiment in our Discussion and Results sections.